# Social distancing in America: Understanding long-term adherence to COVID-19 mitigation recommendations

Christopher P. Reinders Folmer[1]*, Megan A. Brownlee[1], Adam D. Fine[2], Emmeke B. Kooistra[1], Malouke E. Kuiper[1], Elke H. Olthuis[1], Anne Leonore de Bruijn[1], Benjamin van Rooij[1,3]

1 Amsterdam Law School, Center for Law and Behavior, University of Amsterdam, Amsterdam, The Netherlands, 2 School of Criminology and Criminal Justice, Arizona State University, Phoenix, Arizona, United States of America, 3 School of Law, University of California, Irvine, California, United States of America

* c.p.reindersfolmer@uva.nl

**Data Availability Statement:** All data files and analysis syntax are available from the Figshare

## Abstract

A crucial question in the governance of infectious disease outbreaks is how to ensure that people continue to adhere to mitigation measures for the longer duration. The present paper examines this question by means of a set of cross-sectional studies conducted in the United States during the COVID-19 pandemic, in May, June, and July of 2020. Using stratified samples that mimic the demographic characteristics of the U.S. population, it seeks to understand to what extent Americans continued to adhere to social distancing measures in the period after the first lockdown ended. Moreover, it seeks to uncover which variables sustained (or undermined) adherence across this period. For this purpose, we examined a broad range of factors, relating to people's (1) knowledge and understanding of the mitigation measures, (2) perceptions of their costs and benefits, (3) perceptions of legitimacy and procedural justice, (4) personal factors, (5) social environment, and (6) practical circumstances. Our findings reveal that adherence was chiefly shaped by three major factors: respondents adhered more when they (a) had greater practical capacity to adhere, (b) morally agreed more with the measures, and (c) perceived the virus as a more severe health threat. Adherence was shaped to a lesser extent by impulsivity, knowledge of social distancing measures, opportunities for violating, personal costs, and descriptive social norms. The results also reveal, however, that adherence declined across this period, which was partly explained by changes in people's moral alignment, threat perceptions, knowledge, and perceived social norms. These findings show that adherence originates from a broad range of factors that develop dynamically across time. Practically these insights help to improve pandemic governance, as well as contributing theoretically to the study of compliance and the way that rules come to shape behavior.

database (accession number 13125206) https://uvaauas.figshare.com/articles/dataset/Social_Distancing_in_America_Compliance_with_COVID-19_mitigation_measures_in_the_United_States/13125206.

**Funding:** This research was funded by the Dutch Research Council (NWO, https://www.nwo.nl/en) by means of a Corona: Fast-track data grant, awarded to Benjamin van Rooij (grant number 440.20.033).

**Competing interests:** The authors have declared that no competing interests exist.

## Introduction

The global COVID-19 outbreak in 2020 has made clear that the initial defense against a new deadly infectious disease requires large scale behavioral modification. Until there is a vaccine or a cure that can halt a pandemic outbreak, the only protection that people have is to ensure that the spread of the disease is minimized. This entails a range of changes in basic human conduct, from things that have limited economic and social consequences, such as better hand hygiene and the adoption of face masks, to profound, costly changes such as social distancing, forced isolation, quarantine, and broader lockdowns. Such measures only work, however, if people effectively follow them. In this way, the 2020 pandemic has shown the importance of understanding compliance and adherence to outbreak mitigation measures.

There is now a quite well-developed body of research about what made people across the globe follow mitigation measures when they were first adopted. When many governments adopted lockdown rules and social distancing measures as compulsory mandates during this initial "first wave" period, compliance levels were high. This is demonstrated not only by drastic reductions in mobility [1], but also by consequences associated with this, such as the unprecedented event that the price of oil turned negative [2]. A recent review identifies a range of variables that predicted compliance with social distancing measures during the first pandemic wave, including psychosocial, institutional, and situational variables, as well as incentives [3]. Furthermore, this review showed that some highly important policy variables were not associated with compliance during this period. These included for instance deterrence where neither the threat of stricter punishment nor more certain punishment predicted compliance.

After the first wave, many countries lifted the most invasive restrictions, such as lockdowns, and even some of the social distancing measures. Yet as the outbreak was neither controlled nor overcome through a vaccine or medicine, mitigation measures have remained essential for keeping the virus at bay. During the fall, however, many countries found themselves faced with a second pandemic wave. This raises the question of how adherence to mitigation measures has developed during the summer months after the initial strict behavioral measures of the first wave were repealed. Is it the case that social distancing has degraded back toward pre-pandemic normality, and thus gave fertile ground for a resurgence of infections? And if so, which factors shaped such changes and caused people to abandon (or sustain) social distancing?

To understand these questions, the present research collected three cross-sectional surveys in the United States in May, June, and July of 2020. Using stratified samples that mimic the demographic characteristics of the U.S. population, we examined how Americans' adherence to social distancing measures has developed across this period, and which factors have sustained or undermined this. To answer this question, we consider a broad range of influences, which can be arranged into six categories. First, factors related to people's *practical understanding and knowledge* of the measures. Second, factors related to their perception of the *costs and benefits* of the measures. Third, factors related to their perceptions of the *legitimacy and procedural justice* of the measures and the responsible authorities. Fourth, *personal factors* relevant to adherence. Fifth, influences from people's *social environment*. And sixth, *practical circumstances* that may constrain or facilitate their adherence. The paper allows us to understand how these variables shape adherence to social distancing measures in the critical period that follows as a country starts to reopen following a first wave in a pandemic outbreak. By doing so, we contribute to the overall understanding of pandemic governance, as well as to insight into the interaction between rules and human conduct most generally. We also contribute to compliance theory by illuminating how influences at each of these levels may shape adherence

over a longer time period. And finally, we identify important avenues for policy, on how adherence to mitigation measures can be promoted when strict measures are lifted.

## The present study

Following the initial lockdown period in the spring of 2020, the United States underwent dramatic changes, both in terms of the spread of the virus and the measures to counter it. At the beginning of April, approximately 70% of Americans were subject to stay-at-home and social distancing measures [4, 5]. However, by the end of April, infections began to decline [6], and some states began re-opening or reopened altogether, starting with the Southern and Midwestern regions [7]. During the same period, Federal social distancing guidelines were repealed [8], although the requirement remained in place at nearly every state level [9]. Infection rates strongly accelerated from mid-June to late July, however, reaching a peak of almost 75,000 new cases per day [6].

The period between May and July was also characterized by increasing controversy over mitigation measures. There was a continuation of protests against mitigation measures, where people deliberately violated social distancing and other mitigation measures [10, 11]. Furthermore, mitigation measures became increasingly politicized. Compared to Democrats, Republicans voiced greater concern over the economic costs of mitigation measures, and less concern over the threat of the virus [12, 13]. This was illustrated during the 2020 presidential election campaign, where Republican mass rallies were held and some organizers actively countered social distancing measures (e.g., by removing "do not sit here" stickers) [14].

Throughout this period, mitigation measures have remained essential for keeping the virus at bay. But to what extent have Americans followed these measures, and what factors influenced them to do so (or not)? To answer these questions, we leveraged three surveys, collected in May, June, and July of 2020, among stratified samples that mimic the demographic characteristics of the U.S. population.

Our surveys focus on adherence to social distancing recommendations. Although they became less visible in federal public health recommendations after this period, social distancing recommendations continue to exist nearly everywhere at the state level [9]. Our surveys assessed self-reported adherence to social distancing recommendations across various situations, and examine how this has developed in the period after the first wave lockdown. Furthermore, we explored a range of factors that may explain why people did, or did not adhere to these measures, derived from insights on compliance from psychology, criminology, sociology, and economics [5, 15–18]. In operationalizing the present study we broadly distinguish six categories of variables:

1. People's *practical understanding and knowledge* of mitigation measures. In order to be able to adhere to mitigation measures, it is necessary that people have sufficient knowledge of what is expected from them [19–21], and that the measures are clear to them [22]. Accordingly, our surveys firstly test people's knowledge of social distancing measures, and the perceived clarity of the mitigation measures to them. Logically, a lack of knowledge about mitigation measures would be expected to reduce adherence, as would lower perceived clarity.

2. The perceived *costs and benefits* of the mitigation measures. According to the rational choice theory of compliance, people's tendency to adhere should decrease as the costs of doing so become larger, and increase as the benefits improve [23, 24]. Our surveys assess different aspects of this. A first aspect is people's perception of the threat of the virus. Mitigation measures become more beneficial if people regard the virus as a severe threat to their

own health and/or that of others. Yet the health risk of COVID-19 varies between individuals [25–27], as do subjective perceptions of this risk [28]. For this reason, we expected that adherence to social distancing measures would be higher among people who perceive the virus as a greater health threat. The second aspect is the cost people personally face due to the mitigation measures. Due to the pandemic and the measures to mitigate it, many Americans have suffered decreases to their income or employment opportunities [29]. We expected that adherence would be lower among people for whom the personal costs of the mitigation measures are greater. The final aspect is fear of punishment (deterrence). Although social distancing measures were not widely enforced in the U.S., sanctions did occur during the first wave lockdown [30]; furthermore, severe sanctions were communicated for other COVID-related violations [31]. Research on perceptual deterrence suggests that subjective perceptions of punishment may also influence compliance [32]. For this reason, we also examined subjective perceptions of punishment for not following social distancing measures, separating punishment certainty and severity–the key dimensions separated by general deterrence theory [33–35]. We expected that adherence would be higher among people who regarded punishment as more certain, and more severe.

3. The *perceived legitimacy and procedural justice* of the mitigation measures and the responsible authorities. As Max Weber has explained: "So far as it is not derived merely from fear or from motives of expediency, a willingness to submit to an order imposed by one man or a small group, always implies a belief in the legitimate authority (Herrschaftsgewalt) of the source imposing it" (see [36] p. 37). Accordingly, we also aimed to capture such legitimacy perceptions in our study. Jackson and Gau [37] describe legitimacy as the property or quality of possessing rightful power and the acceptance of authority. To the extent that the law and legal authority are perceived to be legitimate, people will feel more obligated to obey the law. Individuals judge legal authority to be legitimate to the extent that they embody the values of being appropriate and proper [38, 39]. Our study assesses six core areas of this. First, we assess people's moral alignment with social distancing measures; i.e., the extent to which they agree with the substance of these measures [40, 41]. During the period that preceded our study, there were clear indications that support for mitigation measures differed among Americans [42, 43]. We expected that adherence would be higher among people with greater substantive support for social distancing measures.

A second core area is people's evaluation of the authorities' responses. To study this, we examined whether people found the overall approach taken by authorities to be consistent and adequate. We expected that adherence would be higher among people who evaluated the authorities' approach more favorably. Relatedly, we assessed procedural justice, or people's perceptions of the procedural fairness through which the rules were made and enforced. The more that people see that rules are made and enforced in a procedurally fair manner, the more likely it is that they will see them as legitimate–and the more likely it becomes that they will feel bound to obey such measures and come to comply with them [40, 41, 44].

In the final area, we assessed people's sense of duty to obey the law. Such sense of duty is a core expression or a downstream consequence of their felt legitimacy, as people with a higher sense of legitimacy, in theory, develop more of a sense of a duty to obey rules developed and enforced by authorities they view as legitimate [45]. We have used three measures to capture this. First is the normative obligation to obey the law, which captures people's sense of duty to obey the law out of moral obligation [46]. Second is the non-normative duty to obey the law, which originates in a sense of coercion or fear, where people feel obligated to obey the law out of fear of the authorities [46]. And last is people's obligation to obey the law in general, which

captures the extent to which they feel that they should obey the law regardless of different circumstances [47–49].

4. *Personal factors* relevant to adherence. As the fourth facet of adherence to social distancing measures, we look at personal factors that are relevant for people's stance toward mitigation measures, or for compliance more generally. A first factor is people's trust in science. Scientific evidence (and indeed, scientists) have played an important (and very visible) role in the public health response to COVID-19, and the measures to mitigate it. Yet trust in science varies between individuals, which may strongly affect their willingness to follow these measures [50, 51]. We expected that adherence to social distancing measures would be higher among people who have greater trust in science. A second, related factor is trust in traditional media. Research suggests that distrust in traditional media is associated with greater belief in misinformation about COVID-19 [52]. This, in turn, predicts lower adherence to measures to mitigate it [53]. Accordingly, we expected adherence to social distancing measures to be higher among people with more trust in traditional media. The third personal factor is impulsivity. To effectively distance oneself from others, it is necessary to inhibit one's usual tendency to get close to them. However, people differ in their capacity to control their impulses, and high levels of impulsivity predict deviant and rule breaking behavior [54–58]. We therefore expected adherence to be lower among more impulsive individuals. Last, we examined people's emotional state. According to strain theory, people may cope with negative emotions through rule violating behavior [59–65]. Indeed, also in context of the COVID-19 pandemic, studies show that negative emotions may lead to lower compliance with quarantine measures [66]. Thus, we expected adherence to be lower among people who experienced more negative emotions.

5. People's *social environment*. As the fifth facet of adherence, we look at influences from people's social environment–specifically descriptive social norms for adhering. In many situations, it is highly visible whether others do (or do not) adhere to social distancing measures. Research shows that perceptions of the norms for complying with particular rules or requests can have an important effect on people's own tendency to do so: the more that they see others comply, the more likely they are to do so themselves; the more that they see others violate or disobey, the more likely they are to offend [67–70]. In light of this, our surveys assessed people's perceptions of the norms for social distancing within their community. We expected adherence to be higher among people who perceived more adherence within their community.

6. People's *practical circumstances*. As the final facet, we looked at the practical circumstances that may shape people's adherence. Whether people can adhere to social distancing measures (or conversely, can violate these) may also be contingent on the extent to which their practical circumstances allow them to do so. Our surveys looked at different aspects of this. First, people's practical capacity to adhere. In order for people to effectively do as social distancing measures demand, it is necessary that their practical circumstances effectively allow them to do so. However, in practice, their capacity to adhere may often vary. For example, keeping a safe distance from others may be more difficult in crowded or constrained environments, or in occupations that cannot be conducted from home or at a distance. Capacity thus may strongly shape adherence, but it should be understood that these concepts are not identical. Simply having the capacity to commit a crime does not mean that one also will do so. The same applies to social distancing: being practically able to keep a distance from others does not mean that someone wishes to do so. We

expected adherence with social distancing measures to be higher among people who had greater practical capacity to adhere to these measures. The second aspect is people's opportunities for violating the measures. In order to violate social distancing recommendations, it first is necessary that there are practical opportunities to do so. However, practical circumstances may make this impossible, for example, when physical environments have been rearranged to separate people from each other. Insights from routine activities theory [71–73] and situational crime prevention [74, 75] show that there is less rule breaking when there are less practical opportunities to do so. We expected greater adherence with social distancing measures among people who saw less opportunities for violating such measures by getting close to others.

## Method

We obtained ethical approval for this project from the Institutional Review Board of the University of California, Irvine and by the Ethics Review Board of the University of Amsterdam. All participants provided consent before participating in the study. Participation was voluntary, and all participants could stop the survey at any time.

### Participants

Participants were residents (18 years or older) of the U.S. that were recruited via the online survey platform SurveyMonkey (https://surveymonkey.com). They were recruited using a stratified sampling approach, in which the final intended sample size was divided into subgroups with the same demographic proportions (age, gender, and race/ethnicity) as the national population based on estimates from the U.S. Census Bureau (https://www.census.gov/). This stratified sampling approach mimics the demographic characteristics of the United States, though it retains the biases and characteristics of a non-probability convenience sample. Three cross-sectional surveys were administered in May, June, and July 2020, using different samples of participants. Participants were paid $3.00 for participating.

1,452 participants took part in Survey 1 (May 8–18). Here, 436 participants were excluded from the sample because they failed to complete the survey, provided incomplete responses, or failed to pass two attention checks. Six participants indicated a nonbinary gender identity; as this number was insufficient for analysis, they were also omitted. The final sample for Survey 1 consisted of 1,012 cases (56.5% women, 43.5% men; $M_{age}$ = 40.32 years).

1,711 participants took part in Survey 2 (June 8–16). Here, 723 participants failed to complete the survey, provided incomplete responses, or failed to pass two attention checks; these participants were excluded. Additionally, five nonbinary participants were omitted from the sample. The final sample for Survey 2 consisted of 986 cases (54.3% women, 45.7% men; $M_{age}$ = 40.17 years).

1,758 participants took part in Survey 3 (July 11–17). Here, 835 participants failed to complete the survey, provided partial responses, or failed to pass two attention checks; again, these participants were excluded. Four nonbinary participants were also omitted. As such, the final sample for Survey 3 consisted of 921 cases (52.7% women, 47.3% men; $M_{age}$ = 40.17 years).

In total, the sample thus consisted of 2,919 cases across three cross-sectional survey waves (54.5% women, 45.5% men; $M_{age}$ = 40.22 years). The sample thus was slightly more female and older than the general population (2019 census: 50.9% women, 49.0% men; $M_{age}$ = 38.3 years) [76]. There was some variability between waves on specific variables (i.e., education, COVID care, inclusion in an ethnic minority group, insurance status, socio-economic status change, and health risk to self and others). These variables were either unrelated to adherence or

**Table 1. Sample characteristics and control variables, Surveys 1 (May), 2 (June), and 3 (July), and full sample.**

|  | Survey 1 (May 8–18) | Survey 2 (June 8–16) | Survey 3 (July 11–17) | Full sample |
|---|---|---|---|---|
| **Age** | 40.29 (12.88) | 40.22 (13.41) | 40.17 (12.87) | 40,22 (13,05) |
| **Gender** |  |  |  |  |
| *Female* | 56.5% | 54.3% | 52.7% | 54,5% |
| *Male* | 43.5% | 45.7% | 47.3% | 45,5% |
| **Region** |  |  |  |  |
| *Northeast* | 20.2% | 20.6% | 20.5% | 20.4% |
| *Midwest* | 21.3% | 19.7% | 21.3% | 20.8% |
| *South* | 44.3% | 42.5% | 41.5% | 42.8% |
| *West* | 14.2% | 17.2% | 16.7% | 16.0% |
| **Minority** | 31.0% | 38.5% | 33.3% | 34.3% |
| **Education** |  |  |  |  |
| *No diploma* | 2.5% | 2.9% | 3.3% | 2.9% |
| *High school degree* | 41.2% | 43.2% | 46.1% | 43.4% |
| *Associate degree* | 12.7% | 13.2% | 13.0% | 13.0% |
| *College degree and higher* | 43.6% | 40.7% | 37.6% | 40.7% |
| **Employed** | 65.7% | 64.0% | 61.8% | 63.9% |
| **Insurance** |  |  |  |  |
| *Uninsured* | 12.9% | 14.9% | 13.6% | 13.8% |
| *Public insurance* | 27.4% | 27.3% | 33.9% | 29.4% |
| *Private insurance* | 59.7% | 57.8% | 52.6% | 56.8% |
| **Socio-econ status, pre-COVID** | 6.05 (1.95) | 6.00 (2.10) | 5.86 (2.10) | 5.97 (2.05) |
| **Socio-econ status, post-COVID** | 5.61 (2.11) | 5.80 (2.20) | 5.63 (2.28) | 5.68 (2.20) |
| **Socio-econ status, change** | -.44 (1.66) | -.20 (1.59) | -.23 (1.70) | -.29 (1.65) |
| **Political orientation** |  |  |  |  |
| *Very progressive* | 16.0% | 20.6% | 17.5% | 18.0% |
| *Slightly progressive* | 25.2% | 24.9% | 24.1% | 24.8% |
| *Slightly conservative* | 29.6% | 28.9% | 27.9% | 28.8% |
| *Very conservative* | 16.7% | 14.8% | 17.7% | 16.4% |
| *Prefer not to say* | 12.4% | 10.7% | 12.8% | 12.0% |
| **Care professionally for COVID** | 6.8% | 10.1% | 9.4% | 8.8% |
| **Health risk self** | 31.9% | 32.4% | 37.9% | 33.9% |
| **Health risk others** | 57.9% | 55.3% | 62.2% | 58.4% |
| **N** | 1012 | 986 | 921 | 2919 |

*Note.* Standard deviations between parentheses.

controlled for in the analyses. Demographical information for all three survey waves and for the full sample is displayed in Table 1.

## Materials

### Survey

Our survey (see Supporting Information) was based on our prior surveys conducted in April 2020 in the United States [5], the United Kingdom [77], the Netherlands [78], and Israel [79]. It assessed the same variables and relied on the same measures. Measures that displayed poor internal consistency in the previous surveys were revised to improve their internal consistency (e.g., adherence, social norms, capacity to adhere, and opportunity to violate); reliability of the

revised measures was high ($\alpha \geq .85$, more details below). Throughout the survey, we referred to COVID-19 as "the coronavirus," which reflects the greater usage of this name in everyday speech, especially during the early stages of the pandemic.

### Control variables

The following demographic variables were recorded: age, gender, nationality, information on residency (state), inclusion in an ethnic minority group, education, employment status, insurance status, social economic status before and after COVID-19 (MacArthur Scale of Subjective Social Status [80]), and political orientation (adapted from [81–83]). For political orientation, a considerable number of participants preferred to not disclose their preference (Survey 1: 12.4%; Survey 2: 10.7%; Survey 3: 12.8%). To enable such cases to be retained in the analysis, this variable was recoded into two dummy variables: one comparing conservative to progressive orientation (1 = very conservative or conservative, 0 = progressive, very progressive, or prefer not to say) and one comparing undisclosed to progressive orientation (1 = prefer not to say, 0 = very conservative, conservative, progressive, very progressive). This approach yielded the same results for adherence as the scale measure, but allowed all cases to be utilized.

Additionally, we asked several questions that probed exposure to and risk from COVID-19. Specifically, we asked participants to indicate whether they provided professional care for coronavirus patients, and whether they or anyone they knew had underlying health issues that would put them more at-risk to suffer complications from the coronavirus.

Correlations between the control variables for all three surveys are displayed in S1–S3 Tables.

### Adherence to social distancing measures

To assess adherence to social distancing measures, we measured participants' self-reported tendency to keep a safe distance from others in various situations [18]. Specifically, we included seven questions that measured their tendency to keep a safe distance (six feet or more) from: (1) "others outside of my direct household," (2) "my neighbors," (3) "colleagues at work," (4) "friends and family from outside of my direct household," (5) "others when grocery shopping," (6) "others when taking a walk or exercising," and (7) "others when commuting or traveling" (1 = "never," 7 = "always"). Responses were mean-scored into a single measure for each wave (Survey 1: $\alpha = .92$; Survey 2: $\alpha = .92$; Survey 3: $\alpha = .93$), with higher scores indicating greater adherence to COVID-19 social distancing measures (see Table 2).

### Practical knowledge and understanding

To assess participants' knowledge and understanding of the mitigation measures, two variables were measured: (1) *knowledge* of these measures, and (2) perceived *clarity* of these measures.

To measure participants' knowledge of mitigation measures, we asked them to indicate whether current COVID-19 mitigation measures required them to keep a safe distance (six feet or more) from others (1 = yes, 2 = no, 3 = don't know). The key comparison is whether people who know that they are under social distancing measures adhere more to these recommendations than people who do not, or are unsure of this. To capture this, these responses were recoded (1 = yes, 0 = no or don't know).

To measure the perceived clarity of mitigation measures, one item was solicited. This asked them to evaluate how clear the measures were that were taken by the authorities to reduce the spread of the coronavirus (1 = "extremely unclear;" 7 = "extremely clear").

**Table 2. Descriptive statistics of dependent variables, Surveys 1 (May), 2 (June), and 3 (July), and full sample.**

|  | Survey 1 | Survey 2 | Survey 3 | Full sample |
|---|---|---|---|---|
|  | (May 8–18) | (June 8–16) | (July 11–17) |  |
| *I keep a safe distance (six feet or more) from*... |  |  |  |  |
| *Others outside of household* | 6.02 (1.41) | 5.85 (1.51) | 5.83 (1.55) | 5.90 (1.49) |
| *Neighbors* | 6.13 (1.36) | 5.85 (1.52) | 5.84 (1.64) | 5.94 (1.51) |
| *Colleagues at work* | 5.88 (1.70) | 5.59 (1.84) | 5.57 (1.91) | 5.68 (1.82) |
| *Friends and family outside household* | 5.67 (1.60) | 5.38 (1.74) | 5.27 (1.84) | 5.45 (1.73) |
| *Others when grocery shopping* | 6.08 (1.26) | 5.93 (1.37) | 5.94 (1.44) | 5.99 (1.36) |
| *Others when walking or exercising* | 6.13 (1.36) | 5.96 (1.46) | 5.94 (1.55) | 6.01 (1.46) |
| *Others when commuting or traveling* | 6.16 (1.39) | 5.95 (1.53) | 5.94 (1.60) | 6.02 (1.51) |
| **Adherence scale measure** | 6.01 (1.20) | 5.79 (1.29) | 5.76 (1.39) | 5.86 (1.30) |
| **N** | 1012 | 986 | 921 | 2919 |

*Note*. Standard deviations between parentheses.

## Costs and benefits

To assess the costs and benefits of the mitigation measures, four variables were measured: (1) the *perceived health threat* of COVID-19, (2) *personal costs* of the mitigation measures, (3) perceptions of the *certainty of punishment* for not following social distancing measures, and (4) perceptions of the *severity of punishment* for failure to do so.

The perceived health threat of COVID-19 was measured by mean-scoring three items. These asked participants to indicate to what extent they believed the coronavirus to be a major threat to (1) their own health, (2) the health of friends and relatives, and (3) the general health (1 = "strongly disagree," 7 = "strongly agree"). Their answers were combined into a scale measure (Survey 1: $\alpha$ = .91; Survey 2: $\alpha$ = .92; Survey 3: $\alpha$ = .92), with higher scores indicating greater perceived health threat.

Personal costs of COVID-19 mitigation measures were assessed by means of five items. Specifically, we asked participants to indicate how likely it was that they would (1) "lose income," (2) "lose their job," (3) "not be able to work," (4) "not be able to work as effectively as normal," and (5) "experience a negative impact on their social life" as a result of the measures (1 = "extremely unlikely," 7 = "extremely likely"). These were combined into a scale measure of personal costs (Survey 1: $\alpha$ = .86; Survey 2: $\alpha$ = .86; Survey 3: $\alpha$ = .86), with higher scores indicating personal greater costs of the mitigation measures.

Perceptions of punishment certainty for violating social distancing measures were measured with two questions. These assessed the perceived likelihood that the authorities would (1) "find out," and (2) "punish you" if participants would not keep a safe distance (six feet or more) from others (1 = "extremely improbable," 7 = "extremely probable"). Both items were highly correlated (Survey 1: $r$ = .75; Survey 2: $r$ = .75; Survey 3: $r$ = .74), and hence were aggregated into a scale measure, with higher scores indicating greater perceived punishment certainty.

Perceptions of punishment severity were assessed using one item. Participants indicated how much they would "suffer" if the authorities would punish them for not keeping a safe distance (six feet or more) from others (1 = "extreme suffering;" 6 = "no suffering at all"). The item was reverse-coded so that higher scores indicate greater perceived punishment severity.

## Legitimacy, procedural justice, and obligation to obey

Six variables were measured to capture participants' perceptions of the legitimacy of the mitigation measures and the responsible authorities, and their felt obligation to follow them: (1)

their *moral alignment* with social distancing measures, (2) their evaluation of the *authority response* to the pandemic, (3) their *normative obligation to obey* the authorities handling the pandemic, (4) their *non-normative obligation to obey* these authorities, (5) their general *obligation to obey the law*, and (6) their perception of the *procedural fairness* of these authorities when enforcing the measures.

Moral alignment with social distancing measures was measured by asking participants to which extent they "morally believe that people should keep a safe distance from others (six feet or more) in order to contain the coronavirus" (1 = "strongly disagree," 7 = "strongly agree").

Evaluation of the authority response was measured using two items. These asked to which extent participants believed the authorities to have been (1) "consistent," and (2) "adequate" in their response to contain the coronavirus (1 = "strongly disagree," 7 = "strongly agree"). Both items were strongly correlated (Survey 1: $r$ = .81; Survey 2: $r$ = .79; Survey 3: $r$ = .80); accordingly, a scale measure was constructed from their responses, with higher scores indicating more favorable evaluations.

Participants' normative obligation to obey the authorities handling COVID-19 was measured by mean-scoring three items (adapted for this study following [46, 84]): (1) "I feel a moral obligation to obey the authorities handling the coronavirus," (2) "I feel a moral duty to support the decisions of the authorities handling the coronavirus, even if I disagree with them," and (3) "I feel a moral duty to obey the instructions of the authorities handling the coronavirus, even when I don't understand the reasons behind them" (1 = "strongly disagree," 5 = "strongly agree"). Answers were aggregated into a scale measure (Survey 1: $\alpha$ = .87; Survey 2: $\alpha$ = .89; Survey 3: $\alpha$ = .90). Higher scores indicated greater normative obligation to obey.

Participants' non-normative obligation to obey the authorities handling COVID-19 was assessed with three items (again adapted for this study following [46, 84]): (1) "people like me have no choice but to obey the authorities handling the coronavirus," (2) "if you don't do what the authorities handling the coronavirus tell you they will treat you badly," and (3) "I only obey the authorities handling the coronavirus because I am afraid of them" (1 = "strongly disagree," 5 = "strongly agree"). Responses were combined into a scale measure (Survey 1: $\alpha$ = .72; Survey 2: $\alpha$ = .73; Survey 3: $\alpha$ = .70), with higher scores indicating greater non-normative obligation to obey.

Participants' general obligation to obey the law was measured using the 12-item Rule Orientation scale [47]. This instrument assesses the perceived acceptability of breaking legal rules across a range of situations (e.g., when the rule is against one's moral principles; when the rule is not enforced; when others think that breaking the rule is justified, etc.; 1 = "strongly disagree," 7 = "strongly agree"). Responses were mean-scored into a scale measure (Survey 1: $\alpha$ = .94; Survey 2: $\alpha$ = .94; Survey 3: $\alpha$ = .94), with higher scores indicating greater felt obligation to obey the law in general.

Perceptions of the authorities' procedural fairness in enforcing the mitigation measures were measured by means of four items (adapted from [40, 85–87]). These asked to which extent they expected that the authorities would: (1) "treat people with respect," (2) "give a person the chance to tell their side of the story if the person is accused of violating measures to contain the coronavirus," (3) "treat people fairly, despite gender, race, religion, or socioeconomic background," and (4) "be honest in enforcing measures to contain the coronavirus" (1 = "strongly disagree," 7 = "strongly agree"). Responses were aggregated into a scale measure (Survey 1: $\alpha$ = .92; Survey 2: $\alpha$ = .93; Survey 3: $\alpha$ = .92), with higher scores indicating greater perceived procedural fairness.

## Personal factors

Four variables were measured to assess personal factors relevant to adherence: participants' (1) *trust in science*, (2) their *trust in traditional media*, (3) their *impulsivity*, and (4) the *negative emotions* that they experience as a result of the pandemic.

Trust in science was measured by means of four items [88]. Participants indicated to which extent they trusted scientists to (1) "create knowledge that is unbiased and accurate," (2) "create knowledge that is useful," (3) "advise government officials on policy," and (4) "inform the public on important issues" (1 = completely distrust, 5 = completely trust). Their answers were mean-scored into a scale measure (Survey 1: α = .92; Survey 2: α = .92; Survey 3: α = .92), with higher scores indicating greater trust in science.

Trust in media was assessed by means of a single item [5]: "Please indicate how much you trust traditional media (e.g., newspapers, TV news, news apps) to be unbiased and accurate"(1 = completely distrust, 5 = completely trust).

Impulsivity was measured by means of a subset of five items taken from the 8-item impulse control subscale from the Weinberger Adjustment Inventory (WAI; [89]): (1) "I should try harder to control myself when I'm having fun," (2) "I do things without giving them enough thought," (3) "When I'm doing something fun (like partying or acting silly), I tend to get carried away and go too far," (4) "I say the first thing that comes to my mind without thinking enough about it," and (5) "I stop and think things through before I act" (1 = "false," 5 = "true;" last item reverse coded). The last item correlated poorly with the other items, and hence was eliminated. The remaining four items were combined into a scale measure (Survey 1: α = .82; Survey 2: α = .81; Survey 3: α = .82), with higher scores indicating greater impulsivity.

Negative emotional state due to COVID-19 was assessed by means of six items. Participants indicated to what extent the coronavirus made them feel (1) "angry," (2) "scared," (3) "powerless," (4) "depressed," (5) "stressed," and (6) "lonely" (1 = "strongly disagree," 7 = "strongly agree"). Responses were aggregated into a scale measure (Survey 1: α = .89; Survey 2: α = .91; Survey 3: α = .90), with higher scores indicating more negative emotions.

## Social environment

To capture influences from the social environment, one variable was measured: perceived *(descriptive) social norms* for adhering to social distancing measures.

Perceived descriptive social norms regarding safe-distancing measures were measured by means of seven items, based on our measure of reported adherence. Participants were asked whether most people they knew were keeping a safe distance (six feet or more) from: (1) "others outside of their direct household," (2) "their neighbors," (3) "colleagues at work," (4) "friends and family from outside of their direct household," (5) "others when grocery shopping," (6) "others when taking a walk or exercising," and (7) "others in traffic or public transport" (1 = "strongly disagree," 7 = "strongly agree"). Participants' answers were combined into a scale measure (Survey 1: α = .94; Survey 2: α = .95; Survey 3: α = .95), with higher scores indicating greater perceived descriptive social norms for adhering.

## Practical circumstances

To assess practical circumstances, two variables were measured: (1) participants' *practical capacity to adhere* to social distancing measures, and (2) their perceived *opportunity to violate* those measures.

Participants' practical capacity to adhere to social distancing mitigation measures was measured by means of seven items, again based on our measures of reported adherence. Participants were asked whether they were capable of keeping a safe distance (six feet or more) from:

**Table 3. Descriptive statistics of independent variables, Surveys 1 (May), 2 (June), and 3 (July), and full sample.**

| | Survey 1 | Survey 2 | Survey 3 | Full sample |
|---|---|---|---|---|
| | (May 8–18) | (June 8–16) | (July 11–17) | |
| **Practical knowledge and understanding** | | | | |
| *Knowledge of measures* | 90.4% | 82.9% | 86.3% | 86.6% |
| *Clarity of measures* | 5.36 (1.62) | 5.15 (1.74) | 5.03 (1.81) | 5.19 (1.73) |
| **Costs and benefits** | | | | |
| *Perceived health threat* | 5.60 (1.47) | 5.53 (1.56) | 5.74 (1.49) | 5.62 (1.51) |
| *Personal costs* | 4.31 (1.62) | 4.09 (1.66) | 4.15 (1.64) | 4.18 (1.64) |
| *Punishment certainty* | 3.34 (1.76) | 3.19 (1.78) | 3.24 (1.74) | 3.26 (1.76) |
| *Punishment severity* | 3.80 (1.70) | 3.80 (1.73) | 3.89 (1.73) | 3.83 (1.72) |
| **Legitimacy** | | | | |
| *Moral alignment* | 6.21 (1.18) | 6.10 (1.34) | 6.15 (1.36) | 6.15 (1.30) |
| *Authority response* | 4.29 (1.85) | 4.36 (1.84) | 3.81 (1.94) | 4.16 (1.89) |
| *Normative obligation to obey* | 3.97 (0.85) | 3.84 (0.91) | 3.90 (0.93) | 3.90 (0.90) |
| *Non-normative obligation to obey* | 2.95 (0.99) | 2.97 (1.02) | 2.94 (0.98) | 2.95 (1.00) |
| *Obligation to obey the law (general)* | 4.40 (1.46) | 4.29 (1.50) | 4.38 (1.49) | 4.36 (1.48) |
| *Procedural justice of enforcement* | 5.24 (1.51) | 5.06 (1.68) | 5.08 (1.65) | 5.13 (1.61) |
| **Personal factors** | | | | |
| *Trust in science* | 3.89 (0.96) | 3.83 (0.99) | 3.83 (1.00) | 3.85 (0.99) |
| *Trust in media* | 2.92 (1.30) | 2.94 (1.30) | 2.83 (1.34) | 2.90 (1.31) |
| *Impulsivity* | 2.40 (1.10) | 2.52 (1.14) | 2.46 (1.13) | 2.46 (1.12) |
| *Negative emotions* | 4.60 (1.53) | 4.53 (1.61) | 4.63 (1.57) | 4.58 (1.57) |
| **Social environment** | | | | |
| *Descriptive social norms* | *5.46 (1.30)* | *5.21 (1.40)* | *5.08 (1.68)* | *5.25 (1.40)* |
| **Practical circumstances** | | | | |
| *Practical capacity to adhere* | 6.06 (0.94) | 5.97 (0.94) | 5.91 (1.08) | 5.98 (0.99) |
| *Opportunity to violate* | 4.46 (1.78) | 4.70 (1.75) | 4.61 (1.71) | 4.59 (1.75) |
| N | 1012 | 986 | 921 | 2919 |

*Note*. Standard deviations between parentheses.

(1) "others outside of my direct household," (2) "my neighbors," (3) "colleagues at work," (4) "friends and family from outside of my direct household," (5) "others when grocery shopping," (6) "others when taking a walk or exercising," and (7) "others in traffic or public transport" (1 = "strongly disagree," 7 = "strongly agree"). Responses were mean-scored into a single scale measure (Survey 1: $\alpha$ = .87; Survey 2: $\alpha$ = .85; Survey 3: $\alpha$ = .89), with higher scores indicating greater practical capacity to adhere.

Opportunity to violate social distancing measures was measured by means of seven items (again based on our measures of adherence). Participants were asked whether, at the present time, it was still possible for them to come within an unsafe distance (closer than six feet) from: (1) "others outside of my direct household," (2) "my neighbors," (3) "colleagues at work," (4) "friends and family from outside of my direct household," (5) "others when grocery shopping," (6) "others when taking a walk or exercising," and (7) "others in traffic or public transport" (1 = "strongly disagree," 7 = "strongly agree"). Responses were aggregated into a single scale measure (Survey 1: $\alpha$ = .94; Survey 2: $\alpha$ = .94; Survey 3: $\alpha$ = .94), with higher scores indicating greater practical opportunity to violate.

Descriptive statistics of all independent variables are displayed for all three samples in Table 3, and correlations are shown in S4–S6 Tables.

## Analysis plan

Our research focused on five major questions: (1) To what extent have Americans adhered to social distancing measures in the period after the first wave lockdown, between May and July 2020, (2) how have the various predictors that were hypothesized to influence adherence developed during this period, (3) which of these predictors in fact influenced adherence during this period, (4) how has the influence of these predictors on adherence changed across this period, and (5) how do the in- and decreases in the level of these predictors that occurred during this period explain the observed changes in adherence? Accordingly, our analysis consisted of three steps.

To examine the first two questions, we explored how adherence to social distancing measures, as well as the situational and motivational variables that were hypothesized to sustain it, evolved from May to July. To do so, we compare these variables between the three survey waves by means of analyses of covariance (ANCOVA), with parameter estimates with robust standard errors (HC3) to conduct pairwise comparisons between months. To illuminate the strictness with which individuals adhere to social distancing recommendations, we also compare frequencies of full adherence. This approach exploits the notion that anyone who reports anything less than full adherence (7 = "always") in fact admits to not having followed the measures (either occasionally or more frequently); this therefore represents a stricter measure of adherence than the average. We compared the frequency of full adherence (across all seven situations) between survey waves using negative binomial regression; to compare the probability of full adherence within specific situations, logistic regression was utilized. All analyses controlled for all demographic and control variables.

To answer the third question, we examined how adherence to social distancing measures was predicted by the various predictors that were hypothesized to sustain it. To do so, we relied on linear (OLS) regression analyses, in which self-reported adherence to social distancing measures was regressed upon these variables (for a similar approach, see [5]). We estimated a hierarchical model in which the different categories of predictors were added to the model in iterated steps. To examine the fourth question, we reran the final iteration of the model expanded with an interaction term, between one of the predictors and survey wave. Separate models were estimated to test the interaction with survey wave for each of the predictors. All analyses were adjusted for heteroscedasticity using Huber/White robust standard error estimation.

Finally, to examine the fifth question, mediation analyses were conducted. These tested how the effect of survey wave on adherence was explained by its indirect effect on the key predictors that were identified in the final step of the hierarchical regression model.

## Results

### Development of adherence levels, May to July

First, we examined how Americans' relative levels of adherence to social distancing measures developed from May to July by comparing average adherence levels between the surveys.

**Average adherence.** Adherence levels on average as well as by situation are displayed in Fig 1. ANCOVA using parameter estimates with robust standard errors indicated that average levels of adherence among Americans declined from May to June (b = -.23, robust SE = .05, $p$ < .001, Cohen's $d$ = .15), but did not change further from June to July (b = -.01, robust SE = .06, $p$ = .797, Cohen's $d$ = .00). When separating the seven situations, adherence declined from May to June in all situations (outside household: b = -.19, robust SE = .07, $p$ = .004, Cohen's $d$ = .11; neighbors: b = -.29, robust SE = .07, $p$ < .001, Cohen's $d$ = .15; colleagues: b = -.29, robust

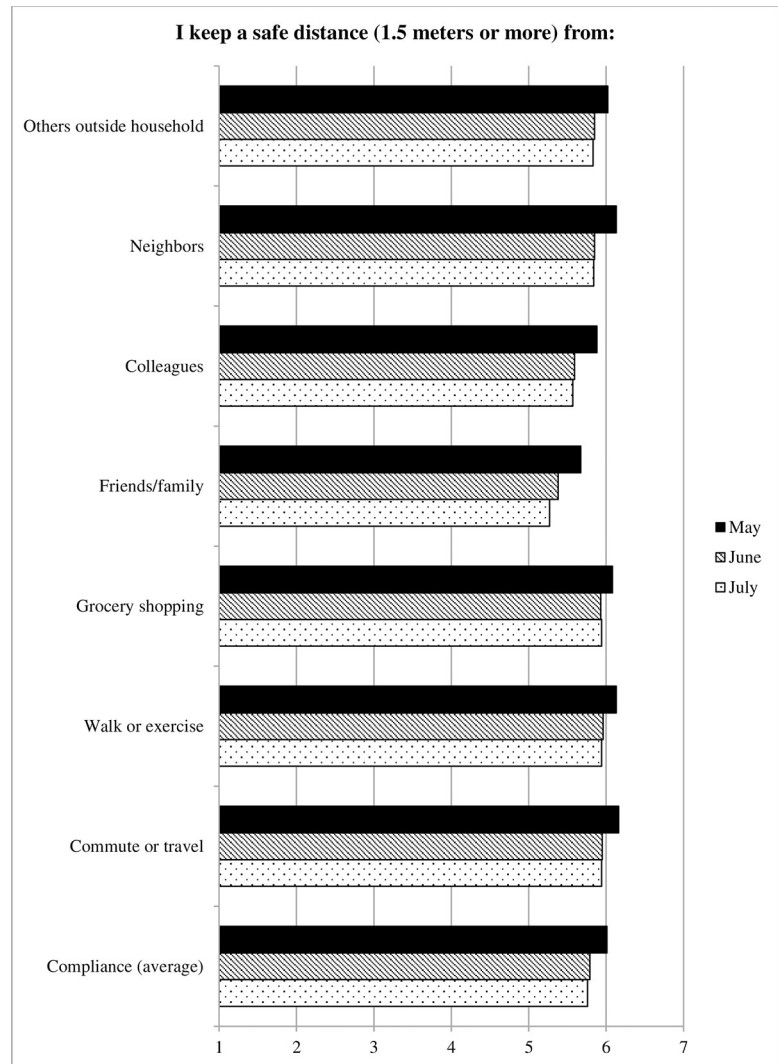

**Fig 1. Adherence to social distancing measures, Survey 1 (May) to Survey 3 (July).**

SE = .08, $p < .001$, Cohen's $d$ = .13; friends and family: b = -.41, robust SE = .08, $p < .001$, Cohen's $d$ = .20; grocery shopping: b = -.13, robust SE = .06, $p$ = .029, Cohen's $d$ = .09; walk or exercise: b = -.18, robust SE = .07, $p$ = .005, Cohen's $d$ = .11; commute or travel: b = -.20, robust SE = .07, $p$ = .003, Cohen's $d$ = .11). From June to July, however, no further significant changes in adherence were observed in any of the situations (all $p$s $\geq$ .269). In sum, the findings suggest a pattern where adherence to social distancing measures declined from May to June (although differences were relatively modest in terms of effect size), but not further in July.

**Full adherence.** Levels of full adherence are displayed in Fig 2. It displays the percentage of participants who reported adhering fully (7 = "always") in each situation (grey and black lines), as well the average percentage of full adherence across all situations (red dashed line). Moreover, it displays the percentage of participants who reported full adherence in all seven situations (red solid lines). When comparing levels of full adherence averaged across all seven situations (red dashed line), negative binomial regression revealed a significant difference between the three survey waves, Wald $\chi^2$ (2) = 13.45, $p$ = .001. Average levels of full adherence

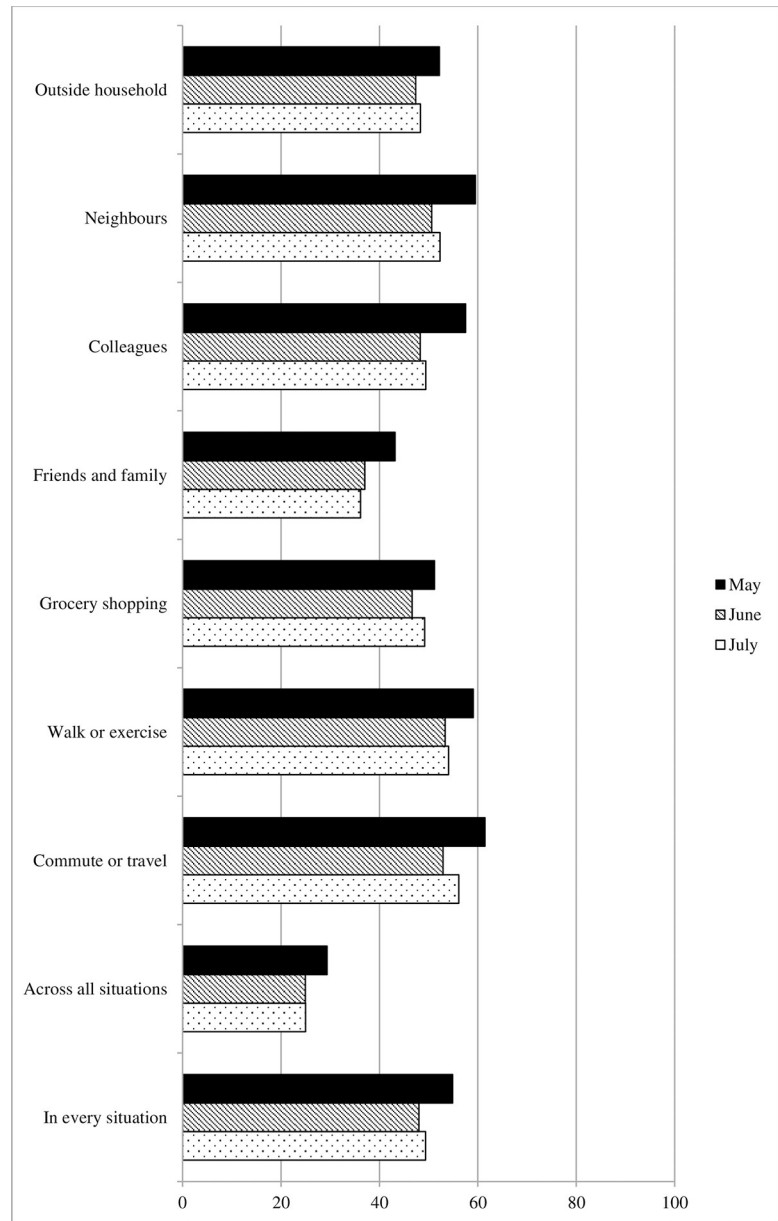

**Fig 2. Full adherence by situation, across all situations, and in every situation, Survey 1 (May) to Survey 3 (July).**

declined by from May to June (b = -.15, SE = .04, Wald $\chi^2$ (1) = 12.16, $p$ < .001 –a reduction of 14.4% relative to May), but did not change further from June to July (b = -.03, SE = .04, Wald $\chi^2$ (1) = 0.57, $p$ = .450). When comparing the number of participants who reported full adherence in every situation (red solid line), there also was a significant decrease from May to June (b = -.27, SE = .10, Wald $\chi^2$ (1) = 6.63, $p$ = .01 –a reduction of 15.0% relative to May). Here also, no further changes were observed from June to July (b = -.02, SE = .11, Wald $\chi^2$ (1) = 0.05, $p$ = .825).

When separating the seven situations (grey and black lines), logistic regression indicated that the probability that participants fully adhered to social distancing recommendation declined significantly from May to June in all situations (outside household: b = -.23, SE = .09,

Wald $\chi^2$ (1) = 6.21, $p$ = .013; neighbors: b = -.39, SE = .09, Wald $\chi^2$ (1) = 17.26, $p$ < .001; colleagues: b = -.39, SE = .09, Wald $\chi^2$ (1) = 17.26, $p$ < .001; friends and family: b = -.32, SE = .09, Wald $\chi^2$ (1) = 11.22, $p$ = .001; grocery shopping: b = -.22, SE = .09, Wald $\chi^2$ (1) = 5.78, $p$ = .016; walk or exercise: b = -.27, SE = .09, Wald $\chi^2$ (1) = 8.20, $p$ = .004; commute or travel: b = -.37, SE = .09, Wald $\chi^2$ (1) = 15.49, $p$ < .001). From June to July, however, probabilities of full adherence did not change any further (all $ps \geq$ .116).

### Development of predictor variables, May to July

**Practical knowledge and understanding.**   Fig 3 displays the development of participants' knowledge of social distancing measures across the three surveys, as well as that of their perceptions of the clarity of those measures. Logistic regression indicated that levels of knowledge of social distancing measures (Table 3) declined significantly in June (b = -.71, SE = .14, Wald $\chi^2$ (1) = 26.03, $p$ < .001), but partially recovered in July (b = .28, SE = .13, Wald $\chi^2$ (1) = 4.48, $p$ = .034). Furthermore, ANCOVA using parameter estimates with robust standard errors indicated that relative to May, the perceived clarity of mitigation measures was significantly lower in July (b = -.34, robust SE = .08, $p$ < .001, Cohen's $d$ = .17).

**Costs and benefits.**   Fig 4 displays the development of the variables reflecting costs and benefits of mitigation measures across the three surveys. Threat perceptions did not change significantly between May and June ($p$ = .073), but increased significantly from June to July (b = .20, robust SE = .07, $p$ = .002, Cohen's $d$ = .11). Conversely, reported personal costs of mitigation measures decreased from May to June (b = -.22, robust SE = .07, $p$ = .002, Cohen's $d$ = .11), as did perceptions of the certainty of punishment (b = -.24, robust SE = .07, $p$ = .002, Cohen's $d$ = .11); neither changed significantly thereafter (both $ps \geq$ .392). Perceptions of the severity of punishment did not change significantly between May and July (all $ps \geq$ .110).

**Legitimacy, procedural justice, and obligation to obey.**   Fig 5 displays the development of the variables reflecting the core constructs in this area. The analyses revealed that moral

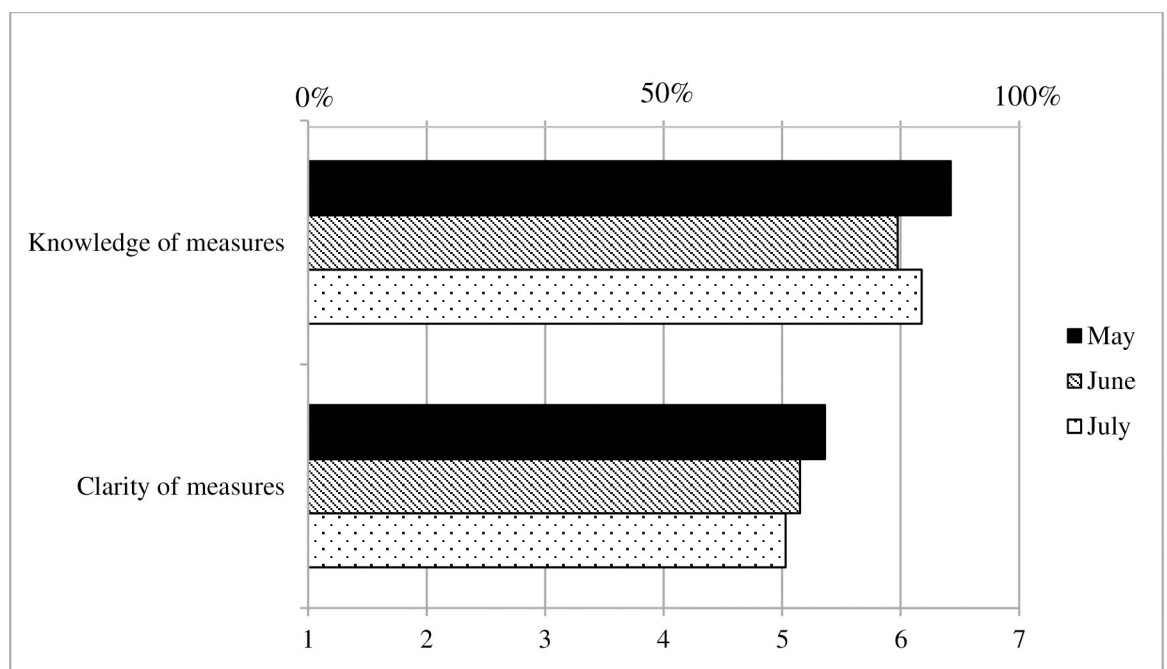

**Fig 3. Practical knowledge and understanding, Survey 1 (May) to Survey 3 (July).**

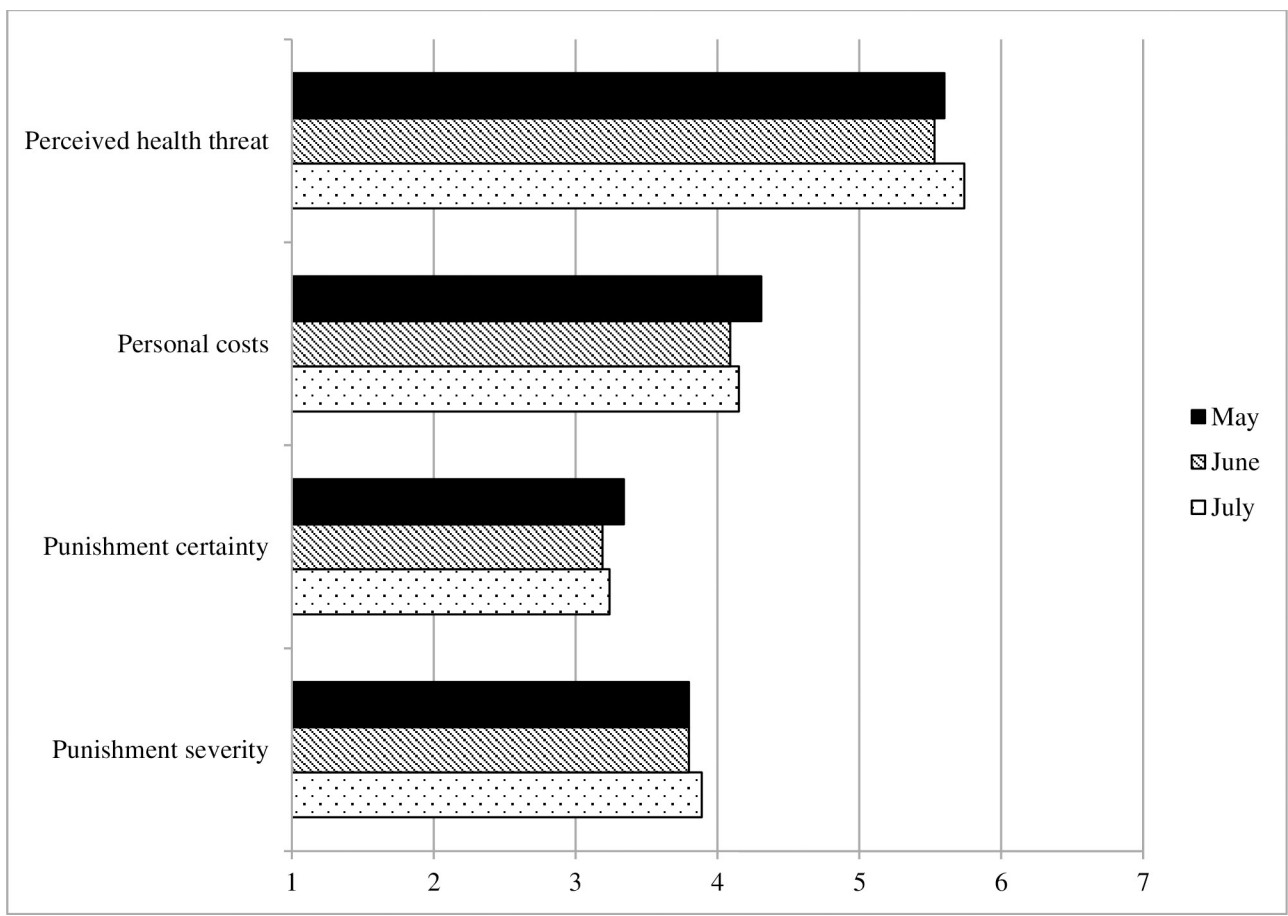

**Fig 4. Costs and benefits of mitigation measures, Survey 1 (May) to Survey 3 (July).**

alignment with social distancing measures declined significantly from May to June (b = -.14, robust SE = .05, $p$ = .011, Cohen's $d$ = .09), while evaluations of the authority response declined significantly from June to July (b = -.54, robust SE = .08, $p$ < .001, Cohen's $d$ = .25). Furthermore, there was a significant decline from May to June in participants' normative obligation to obey the authorities handling COVID-19 (b = -.15, robust SE = .04, $p$ < .001, Cohen's $d$ = .14), and in perceptions of their procedural fairness (b = -.21, robust SE = .07, $p$ = .003, Cohen's $d$ = .11). No significant changes were observed in non-normative obligation to obey these authorities, however, or in their general obligation to obey the law (all $ps \geq$ .214).

**Personal factors.** Fig 6 shows the development of personal factors relevant to adherence. The results revealed small, but significant changes in trust in science and media: trust in science decreased significantly from May to June (b = -.09, robust SE = .04, $p$ = .032, Cohen's $d$ = .09), whereas trust in mainstream media showed a significant decrease from May to July (b = -.12, robust SE = .06, $p$ = .030, Cohen's $d$ = .09). No significant changes were observed in impulsivity (all $ps \geq$ .092) or negative emotions (all $ps \geq$ .554).

**Social environment.** Fig 7 shows the development of perceived (descriptive) social norms for adhering to social distancing measures. From May to July, perceived social norms for keeping a safe distance were significantly reduced (b = -.38, robust SE = .06, $p$ < .001, Cohen's $d$ = .23).

**Practical circumstances.** Finally, Fig 8 displays the development in practical circumstances for adhering. From May to July, there was a significant decrease in respondents'

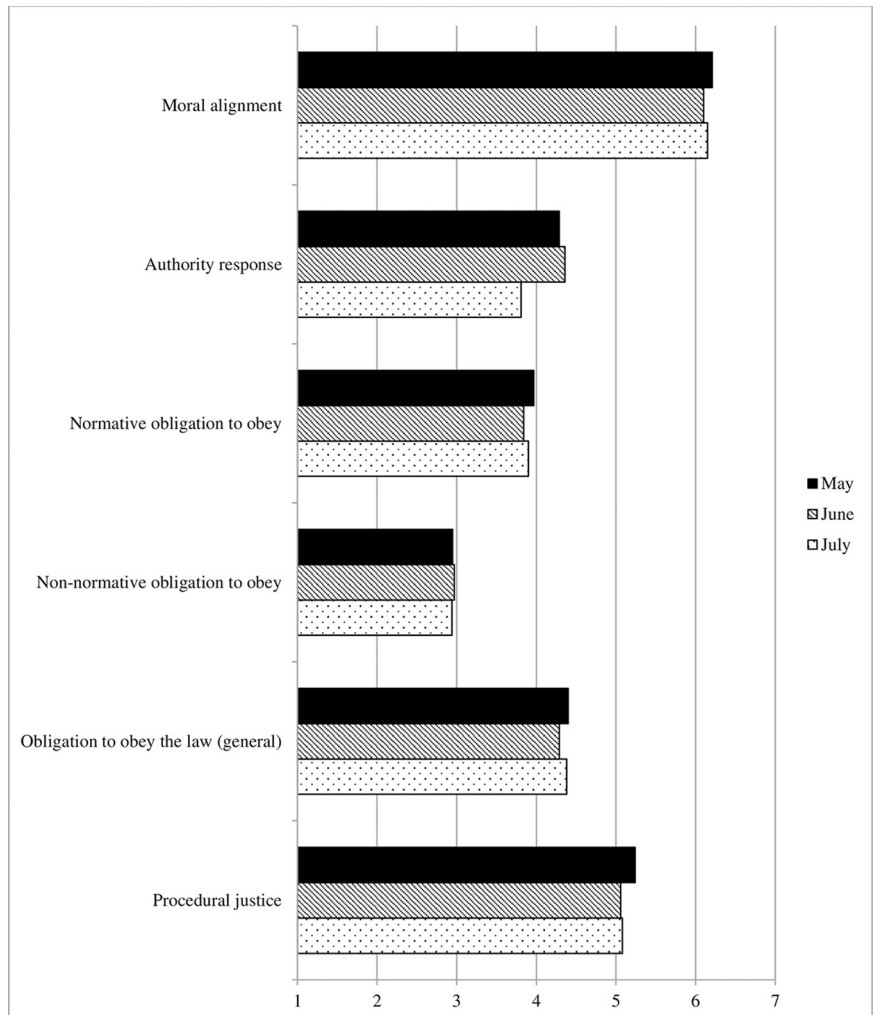

**Fig 5. Legitimacy variables, Survey 1 (May) to Survey 3 (July).**

reported capacity to adhere to social distancing measures (b = -.14, robust SE = .05, $p$ = .002, Cohen's $d$ = .11). Conversely, perceived opportunities for violate social distancing measures became significantly greater from May to June (b = .20, robust SE = .08, $p$ = .013, Cohen's $d$ = .09).

## Understanding adherence to social distancing measures from May to July

**Hierarchical regression model.** As the previous section demonstrates, adherence to social distancing measures declined significantly in the period after the initial first wave lockdown. At the same time, significant changes were observed in many of the variables that were hypothesized to shape adherence. Our next major question is to understand how these processes shaped adherence to social distancing measures during this period. To do so, we estimated a linear regression model, in which adherence was regressed upon the various predictors in a series of hierarchical steps. This model was estimated using the combined data from all three survey waves ($N$ = 2,919), with survey wave included as an additional predictor (1 = May, 2 = June, 3 = July). Collinearity statistics indicated no issues with multicollinearity (all VIFs ≤ 2.55; all tolerances ≥ .39). Table 4 displays the results.

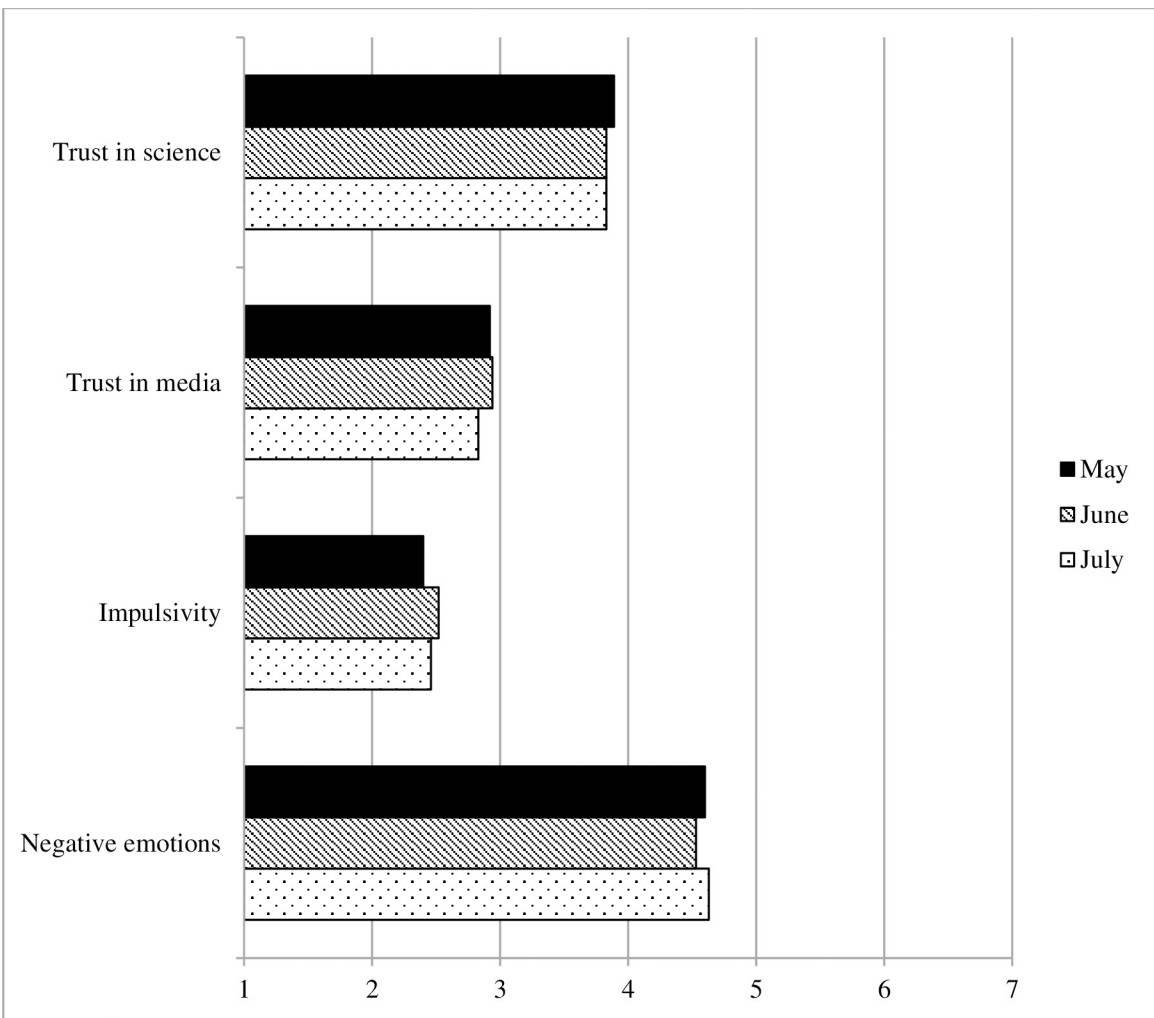

**Fig 6. Personal factors, Survey 1 (May) to Survey 3 (July).**

In Step 1, the model included only the survey wave dummies and the control variables. As shown in Table 4 (column 1), relative to May, adherence decreased significantly in June and in July. Adherence was significantly higher among older participants, female participants,

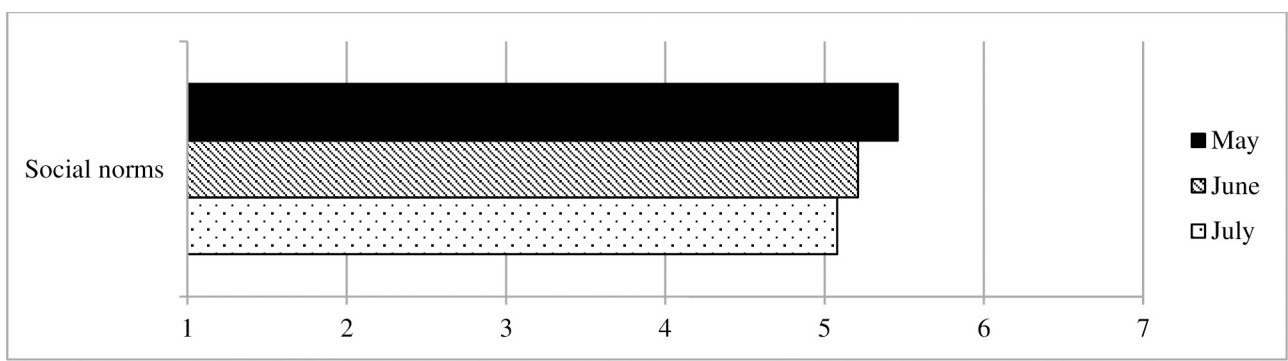

**Fig 7. Social environment, Survey 1 (May) to Survey 3 (July).**

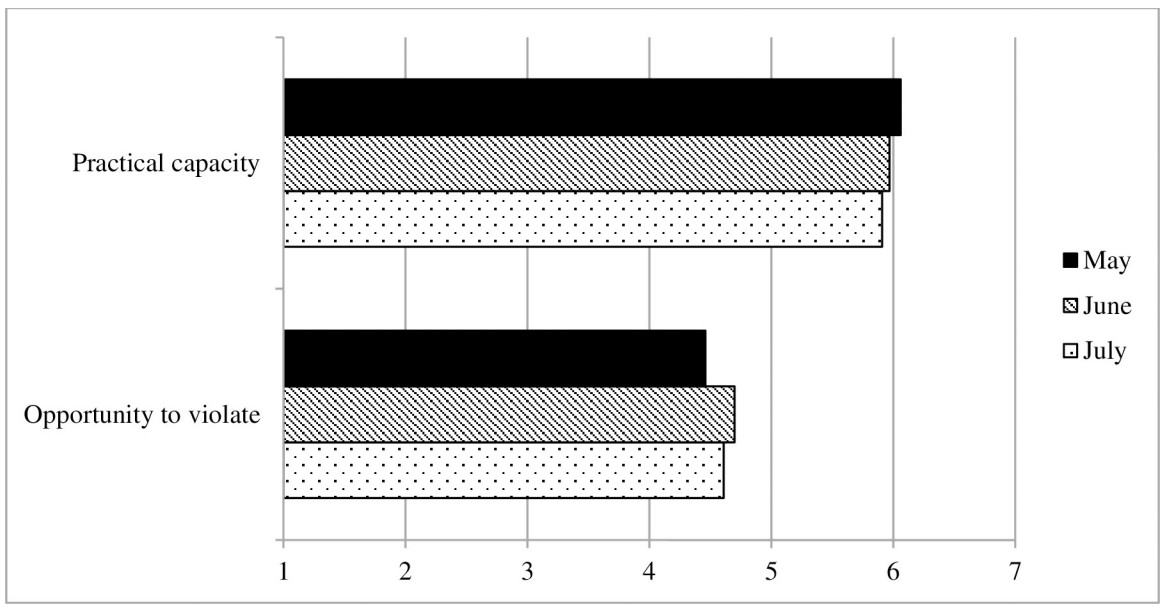

**Fig 8. Practical circumstances, Survey 1 (May) to Survey 3 (July).**

minority group members, people with higher levels of education, and people with higher socio-economic status. Furthermore, adherence was greater among people who suffered from a health condition that placed them at increased risk, or who knew others who suffered from such health conditions. Conversely, adherence was significantly lower among people who professionally cared for COVID patients, and among participants with more conservative political orientations. Lastly, relative to participants from the Northeast region, adherence was significantly lower among participants from the Midwest and the South regions. Together, the model explained only 8% of the variability in adherence, however.

In Step 2, predictors reflecting participants' practical knowledge and understanding of the mitigation measures were added to the model. Adherence was significantly greater among participants who had greater knowledge of social distancing measures, and among participants who regarded these measures as more clear. Inclusion of these predictors meant that the effect of health risk to others was no longer significant. The model now explained 14% of the variance in adherence, a significant increase relative to Step 1 ($\chi^2$ (2) = 223.87, $p < .001$).

In Step 3, predictors related to the costs and benefits of adherence were entered into the model. Adherence was significantly greater among participants and who regarded the COVID-19 pandemic as more threatening. Also, adherence was greater among participants for whom the costs of adhering was higher. Conversely, punishment perceptions did not predict adherence. Inclusion of these variables rendered the effect of political orientation nonsignificant. The model explained 31% of the variance in adherence, a significant increase over Step 2 ($\chi^2$ (4) = 622.75, $p < .001$).

In Step 4, predictors reflecting participants' legitimacy perceptions were added to the model. Adherence was significantly higher among participants who morally agreed more with the measures (i.e., moral alignment), who felt greater normative obligation to obey the COVID-19 authorities, and who felt a higher general obligation to obey the law. Inclusion of these variables rendered nonsignificant the effect of perceived clarity of mitigation measures, socio-economic status, and Midwest region. The model explained 39% of the variance in adherence, a significant increase over Step 3 ($\chi^2$ (5) = 383.19, $p < .001$).

**Table 4. Hierarchical linear regression (with robust standard errors), adherence to mitigation measures by predictor and control variables.**

| | Step 1 | Step 2 | Step 3 | Step 4 | Step 5 | Step 6 | Step 7 | Effect size (Cohen's d) |
|---|---|---|---|---|---|---|---|---|
| **Survey wave** | | | | | | | | |
| *Month: June (vs May)* | -.23*** (.05) | -.14** (.05) | -.13** (.05) | -.10* (.05) | -.09* (.04) | -.06 (.04) | -.09* (.04) | .08 |
| *Month: July (vs May)* | -.24*** (.06) | -.17** (.06) | -.23*** (.05) | -.20*** (.05) | -.19*** (.05) | -.15** (.05) | -.14** (.04) | .12 |
| **Control variables** | | | | | | | | |
| *Age* | .01*** (.00) | .01*** (.00) | .01*** (.00) | .01*** (.00) | .00** (.00) | .00** (.00) | .00* (.00) | .09 |
| *Gender, female (vs male)* | .26*** (.05) | .23*** (.05) | .20*** (.04) | .15*** (.04) | .13** (.04) | .13** (.04) | .12** (.03) | .13 |
| *Minority* | .18*** (.05) | .12* (.05) | -.01 (.04) | -.01 (.04) | -.00 (.04) | -.01 (.04) | -.00 (.04) | .00 |
| *Education* | .05** (.02) | .06*** (.02) | .06*** (.01) | .05** (.01) | .04** (.01) | .04** (.01) | .03** (.01) | .11 |
| *Employed* | -.06 (.05) | -.06 (.05) | -.05 (.05) | -.02 (.04) | -.02 (.04) | -.02 (.04) | .01 (.04) | .01 |
| *COVID Care* | -.22* (.09) | -.26** (.09) | -.33*** (.08) | -.23** (.08) | -.18* (.08) | -.16* (.07) | -.05 (.07) | .03 |
| *Insurance, public (vs no)* | .09 (.08) | .05 (.07) | .08 (.07) | .07 (.07) | .07 (.06) | .07 (.06) | .09 (.06) | .06 |
| *Insurance, private (vs no)* | .12 (.08) | .07 (.08) | .06 (.07) | .06 (.07) | .07 (.07) | .08 (.07) | .13* (.06) | .09 |
| *Socio-economic status, pre-COVID* | .04** (.01) | .03* (.01) | .03* (.01) | .02 (.01) | .02 (.01) | .01 (.01) | .00 (.01) | .02 |
| *Socio-economic status change (post-pre)* | -.00 (.02) | -.01 (.01) | .02 (.01) | .01 (.01) | .01 (.01) | .00 (.01) | .00 (.01) | .00 |
| *Health risk self* | .16** (.05) | .17** (.05) | -.05 (.04) | .02 (.04) | .03 (.04) | .03 (.04) | .03 (.04) | .03 |
| *Health risk others* | .13* (.05) | .08 (.05) | -.02 (.04) | -.03 (.04) | -.04 (.04) | -.03 (.04) | -.02 (.04) | .02 |
| *Political orientation, conservative (vs liberal)* | -.26*** (.05) | -.23*** (.05) | -.01 (.04) | .05 (.04) | .06 (.05) | .05 (.04) | .05 (.04) | .05 |
| *Political orientation, not disclosed (vs liberal)* | -.12 (.08) | -.06 (.08) | .10 (.07) | .13 (.07) | .15* (.07) | .15* (.07) | .11 (.06) | .07 |
| *Region: Midwest (vs Northeast)* | -.30*** (.07) | -.24** (.07) | -.13* (.06) | -.10 (.06) | -.08 (.06) | -.07 (.06) | -.10 (.05) | .07 |
| *Region: South (vs Northeast)* | -.23*** (.06) | -.17** (.06) | -.14* (.05) | -.13** (.05) | -.12* (.05) | -.09 (.05) | -.12** (.04) | .10 |
| *Region: West (vs Northeast)* | -.04 (.07) | .00 (.07) | .01 (.06) | .01 (.06) | .02 (.06) | .02 (.06) | .00 (.05) | .00 |
| **Practical knowledge and understanding** | | | | | | | | |
| *Knowledge of measures* | | .66*** (.09) | .45*** (.07) | .33*** (.07) | .31*** (.07) | .28*** (.07) | .18** (.06) | .13 |
| *Clarity of measures* | | .13*** (.01) | .06*** (.01) | .01 (.01) | .00 (.01) | -.00 (.01) | -.02 (.01) | .06 |
| **Costs and benefits** | | | | | | | | |
| *Perceived health threat* | | | .39*** (.02) | .14*** (.02) | .14*** (.02) | .14*** (.02) | .13*** (.02) | .27 |
| *Personal costs* | | | .03* (.01) | .04** (.01) | .03* (.01) | .03* (.01) | .03* (.01) | .08 |
| *Punishment certainty* | | | -.01 (.01) | .01 (.01) | .02 (.01) | .01 (.01) | .01 (.01) | .02 |
| *Punishment severity* | | | .01 (.01) | -.00 (.01) | -.00 (.01) | -.00 (.01) | -.00 (.01) | .01 |
| **Legitimacy** | | | | | | | | |
| *Moral alignment* | | | | .37*** (.03) | .36*** (.03) | .34*** (.03) | .25*** (.03) | .47 |
| *Authority response* | | | | -.01 (.01) | -.00 (.01) | -.01 (.01) | -.01 (.01) | .03 |
| *Normative obligation to obey* | | | | .11*** (.03) | .10*** (.03) | .09** (.03) | .02 (.03) | .03 |
| *Non-normative obligation to obey* | | | | .01 (.02) | .02 (.02) | -.00 (.02) | .02 (.02) | .03 |
| *Obligation to obey the law (general)* | | | | .05*** (.01) | .02 (.02) | .02 (.02) | .01 (.01) | .04 |
| *Procedural justice of enforcement* | | | | .00 (.01) | .00 (.01) | -.01 (.01) | -.01 (.01) | .03 |
| **Personal factors** | | | | | | | | |
| *Trust in science* | | | | | .08** (.03) | .06* (.03) | .04 (.02) | .06 |
| *Trust in media* | | | | | -.02 (.02) | -.03 (.02) | -.01 (.01) | .03 |
| *Impulsivity* | | | | | -.12*** (.02) | -.13*** (.02) | -.08*** (.02) | .17 |

*(Continued)*

**Table 4.** (Continued)

| | Step 1 | Step 2 | Step 3 | Step 4 | Step 5 | Step 6 | Step 7 | Effect size (Cohen's d) |
|---|---|---|---|---|---|---|---|---|
| *Negative emotions* | | | | | .03 (.01) | .02 (.01) | .02 (.01) | .05 |
| **Social environment** | | | | | | | | |
| *Descriptive social norms* | | | | | | .15*** (.02) | *.03* (.01) | *.07* |
| **Practical circumstances** | | | | | | | | |
| *Practical capacity to adhere* | | | | | | | .52*** (.03) | .89 |
| *Opportunity to violate* | | | | | | | -.03** (.01) | .10 |
| *Constant* | 5.05*** (.16) | 3.98*** (.17) | 2.29*** (.19) | 1.09*** (.20) | 1.25*** (.22) | 1.08*** (.22) | -.10 (.21) | |
| **Rsq** | .08 | .14 | .31 | .39 | .40 | .42 | .52 | |

*Note*. Robust standard errors between parentheses.

* $p < .05$

** $p < .01$

*** $p < .001$.

In Step 5, personal factors were entered into the model. Adherence was significantly higher among participants who had greater trust in science. Conversely, adherence was significantly lower among more impulsive participants. Controlling for these variables rendered nonsignificant the effect of general obligation to obey the law, and revealed a significant effect of undisclosed political orientation, which predicted greater adherence (relative to liberals). The model explained 40% of the variance in adherence, a significant increase over Step 4 ($\chi^2$ (5) = 46.47, $p < .001$).

In Step 6, predictors reflecting the social environment were added to the model. Adherence was significantly higher among people who perceived stronger (descriptive) social norms for keeping a safe distance. Inclusion of this variable rendered nonsignificant the effect of South region. Additionally, the decline in adherence from May to June was reduced to nonsignificance when this variable was included. The model explained 42% of the variance in adherence, a significant increase over Step 5 ($\chi^2$ (1) = 103.62, $p < .001$).

Finally, in Step 7, predictors reflecting the practical circumstances were entered into the model. Adherence was significantly higher among people who had greater practical ability to keep at a safe distance from others. In contrast, adherence was significantly lower among people who saw more opportunities for violating social distancing measures. By including these variables in the model, the effects of trust in science, normative obligation to obey, and care for COVID patients were reduced to nonsignificance. Conversely, inclusion of these variables restored to significance the previously observed effect of South region, and the decline in adherence from May to June. Last, inclusion of these variables revealed a significant effect of private insurance, such that adherence was greater among participants who had private (relative to no) insurance. The final model explained 52% of the variance in adherence, a significant increase over Step 6 ($\chi^2$ (2) = 532.61, $p < .001$).

**Change in predictors across waves.** In addition, we sought to understand how the effect of these predictors on adherence changed across survey waves. To do so, we estimated additional models based on the final step of the hierarchical regression model (step 7, Table 4). Each of these models included all the predictors and control variables from the hierarchical model (as in step 7), and one single interaction term, between survey wave and one of the 19 predictors (respectively knowledge of measures, clarity of measures, perceived health threat, personal costs, punishment certainty, punishment severity, moral alignment, authority

response, normative obligation to obey, non-normative obligation to obey, general obligation to obey the law, procedural justice of enforcement, trust in science, trust in media, impulsivity, negative emotions, descriptive social norms, practical capacity to adhere, or opportunity to violate). In total, 19 interaction models therefore were estimated, each including a single interaction term. Because our interest with these models was exclusively in the interactive effects with survey waves, only the interaction terms are displayed in Table 5, for each of the 19 models.

The results of these analyses indicated that largely, the effect of the predictors on adherence did not vary across waves. Of the key predictors of adherence in the final hierarchical model (i.e., knowledge, perceived threat, personal costs, moral alignment, impulsivity, descriptive social norms, practical capacity to adhere, and opportunity to violate, see Table 4, step 7), only the effect of impulsivity varied across survey waves (in May: b = -.13, $SE$ = .03, $p$ < .001; in July: b = -.06, $SE$ = .03, $p$ = .049; contrast = .07, SE = .04, $p$ = .049). The results did indicate changes across waves in the effects of normative obligation to obey (in May: b = -.06, $SE$ = .04, $p$ = .098; in July: b = .08, $SE$ = .04, $p$ = .066; contrast = .13, SE = .05, $p$ = .006), non-normative obligation to obey (in May: b = -.07, $SE$ = .03, $p$ = .043; in July: b = .10, $SE$ = .03, $p$ = .001; contrast = .17, SE = .04, $p$ < .001), trust in science (in May: b = -.04, $SE$ = .03, $p$ = .181; in July: b = .10, $SE$ = .04, $p$ = .008; contrast = .14, SE = .04, $p$ = .001), and trust in media (in May: b = -.06, $SE$ = .02, $p$ = .011; in July: b = .02, $SE$ = .02, $p$ = .511; contrast = .07, SE = .03, $p$ = .021). Thus, impulsivity became gradually less important as a predictor of adherence across waves, while particularly non-normative obligation to obey and trust in science became more influential. These changes were modest in terms of effect size, however.

Although the effect of the key predictors of adherence generally did not vary across waves (i.e., they did not become more or less predictive of adherence), the absolute levels of these variables did change significantly between May and July. Indeed, as was previously shown in the descriptive analyses of changes across survey waves, there were significant changes during this period in participants' reported knowledge of mitigation measures, their perceptions of the threat of the virus, their personal costs of the mitigation measures, their moral alignment with those measures, their perceived social norms, their practical capacity to comply, and their perceived opportunities for violating social distancing measures. To examine how these changes contributed to the observed decline in compliance across this period, we finally conducted mediation analyses. These tested whether the effect of survey wave on compliance was mediated by the effect of survey wave on each of the key predictors of adherence.

To do so, we relied on the PARAMED module in Stata [90], which can handle both linear and categorical mediators. In these models, survey wave was the independent variable, adherence the dependent variable, and the mediator was either reported knowledge of mitigation measures, perceptions of the threat of the virus, personal costs of the mitigation measures, moral alignment with mitigation measures, perceived social norms, practical capacity to comply, or perceived opportunities for violating. The models controlled for all other predictors and control variables, and featured bias-corrected bootstrap confidence intervals (1,000 replications). In total, 14 mediation models were estimated (for 7 mediators, with two models each: one comparing wave 1 to wave 2, and one comparing wave 1 to wave 3). Results are presented in Table 6.

The indirect effects reported in Table 6 suggest that the effect of survey wave on adherence (the total effect) was significantly reduced, and thus partially mediated (i.e., confidence interval of the indirect effect does not include zero) by the following variables: knowledge of mitigation measures, perceived health threat, moral alignment, and social norms. Conversely, personal costs and capacity to adhere did not mediate this effect. Accordingly, these findings suggest that the observed decrease in adherence from May to July was partially explained by their

**Table 5. Interaction models: Interaction effects on adherence to mitigation measures of predictors by survey waves.**

| | Model 8 | Model 9 | Model 10 | Model 11 | Model 12 | Model 13 | Model 14 | Model 15 | Model 16 | Model 17 | Model 18 | Model 19 | Model 20 | Model 21 | Model 22 | Model 23 | Model 24 | Model 25 | Model 26 | Effect size (Cohen's d) |
|---|---|---|---|---|---|---|---|---|---|---|---|---|---|---|---|---|---|---|---|---|
| **Practical knowledge and understanding** | | | | | | | | | | | | | | | | | | | | |
| *Knowledge of measures* | | | | | | | | | | | | | | | | | | | | |
| *x Month: June (vs May)* | .03 (.15) | | | | | | | | | | | | | | | | | | | .01 |
| *x Month: July (vs May)* | -.07 (.16) | | | | | | | | | | | | | | | | | | | .02 |
| *Clarity of measures* | | | | | | | | | | | | | | | | | | | | |
| *x Month: June (vs May)* | | .04 (.02) | | | | | | | | | | | | | | | | | | .06 |
| *x Month: July (vs May)* | | .03 (.02) | | | | | | | | | | | | | | | | | | .04 |
| **Costs and benefits** | | | | | | | | | | | | | | | | | | | | |
| *Perceived health threat* | | | | | | | | | | | | | | | | | | | | |
| *x Month: June (vs May)* | | | .04 (.03) | | | | | | | | | | | | | | | | | .06 |
| *x Month: July (vs May)* | | | .04 (.03) | | | | | | | | | | | | | | | | | .05 |
| *Personal costs* | | | | | | | | | | | | | | | | | | | | |
| *x Month: June (vs May)* | | | | .00 (.02) | | | | | | | | | | | | | | | | .01 |
| *x Month: July (vs May)* | | | | .03 (.03) | | | | | | | | | | | | | | | | .05 |
| *Punishment certainty* | | | | | | | | | | | | | | | | | | | | |
| *x Month: June (vs May)* | | | | | .01 (.02) | | | | | | | | | | | | | | | .02 |
| *x Month: July (vs May)* | | | | | .01 (.03) | | | | | | | | | | | | | | | .02 |
| *Punishment severity* | | | | | | | | | | | | | | | | | | | | |
| *x Month: June (vs May)* | | | | | | -.01 (.02) | | | | | | | | | | | | | | .02 |
| *x Month: July (vs May)* | | | | | | -.01 (.02) | | | | | | | | | | | | | | .01 |
| **Legitimacy** | | | | | | | | | | | | | | | | | | | | |
| *Moral alignment* | | | | | | | | | | | | | | | | | | | | |
| *x Month: June (vs May)* | | | | | | | .03 (.04) | | | | | | | | | | | | | .03 |
| *x Month: July (vs May)* | | | | | | | .02 (.04) | | | | | | | | | | | | | .03 |
| *Authority response* | | | | | | | | | | | | | | | | | | | | |
| *x Month: June (vs May)* | | | | | | | | .02 (.02) | | | | | | | | | | | | .04 |
| *x Month: July (vs May)* | | | | | | | | .02 (.02) | | | | | | | | | | | | .04 |

(*Continued*)

Table 5. (Continued)

| | Model 8 | Model 9 | Model 10 | Model 11 | Model 12 | Model 13 | Model 14 | Model 15 | Model 16 | Model 17 | Model 18 | Model 19 | Model 20 | Model 21 | Model 22 | Model 23 | Model 24 | Model 25 | Model 26 | Effect size (Cohen's d) |
|---|---|---|---|---|---|---|---|---|---|---|---|---|---|---|---|---|---|---|---|---|
| *Normative obligation to obey* | | | | | | | | | | | | | | | | | | | | |
| *x Month: June (vs May)* | | | | | | | | | .09 (.05) | | | | | | | | | | | .07 |
| *x Month: July (vs May)* | | | | | | | | | .13** (.05) | | | | | | | | | | | .11 |
| *Non-normative obligation to obey* | | | | | | | | | | | | | | | | | | | | |
| *x Month: June (vs May)* | | | | | | | | | | .09* (.04) | | | | | | | | | | .09 |
| *x Month: July (vs May)* | | | | | | | | | | .17*** (.04) | | | | | | | | | | .15 |
| *Obligation to obey the law (general)* | | | | | | | | | | | | | | | | | | | | |
| *x Month: June (vs May)* | | | | | | | | | | | -.02 (.03) | | | | | | | | | .03 |
| *x Month: July (vs May)* | | | | | | | | | | | -.03 (.03) | | | | | | | | | .04 |
| *Procedural justice of enforcement* | | | | | | | | | | | | | | | | | | | | |
| *x Month: June (vs May)* | | | | | | | | | | | | .05 (.03) | | | | | | | | .07 |
| *x Month: July (vs May)* | | | | | | | | | | | | .04 (.03) | | | | | | | | .06 |
| **Personal factors** | | | | | | | | | | | | | | | | | | | | |
| *Trust in science* | | | | | | | | | | | | | | | | | | | | |
| *x Month: June (vs May)* | | | | | | | | | | | | | .10* (.04) | | | | | | | .09 |
| *x Month: July (vs May)* | | | | | | | | | | | | | .14** (.04) | | | | | | | .13 |
| *Trust in media* | | | | | | | | | | | | | | | | | | | | |
| *x Month: June (vs May)* | | | | | | | | | | | | | | .06* (.03) | | | | | | .07 |
| *x Month: July (vs May)* | | | | | | | | | | | | | | .07* (.03) | | | | | | .09 |
| *Impulsivity* | | | | | | | | | | | | | | | | | | | | |
| *x Month: June (vs May)* | | | | | | | | | | | | | | | .07 (.04) | | | | | .07 |
| *x Month: July (vs May)* | | | | | | | | | | | | | | | .07* (.04) | | | | | .07 |
| *Negative emotions* | | | | | | | | | | | | | | | | | | | | |
| *x Month: June (vs May)* | | | | | | | | | | | | | | | | .04 (.03) | | | | .06 |
| *x Month: July (vs May)* | | | | | | | | | | | | | | | | .02 (.03) | | | | .03 |
| *Social environment* | | | | | | | | | | | | | | | | | | | | |
| *Descriptive social norms* | | | | | | | | | | | | | | | | | | | | |

*(Continued)*

**Table 5.** (Continued)

| | Model 8 | Model 9 | Model 10 | Model 11 | Model 12 | Model 13 | Model 14 | Model 15 | Model 16 | Model 17 | Model 18 | Model 19 | Model 20 | Model 21 | Model 22 | Model 23 | Model 24 | Model 25 | Model 26 | Effect size (Cohen's d) |
|---|---|---|---|---|---|---|---|---|---|---|---|---|---|---|---|---|---|---|---|---|
| **x Month: June (vs May)** | | | | | | | | | | | | | | | | | .02 (.03) | | | .02 |
| **x Month: July (vs May)** | | | | | | | | | | | | | | | | | .05 (.03) | | | .07 |
| **Practical circumstances** | | | | | | | | | | | | | | | | | | | | |
| **Practical capacity to adhere** | | | | | | | | | | | | | | | | | | | | |
| **x Month: June (vs May)** | | | | | | | | | | | | | | | | | | .00 (.05) | | .00 |
| **x Month: July (vs May)** | | | | | | | | | | | | | | | | | | .04 (.05) | | .04 |
| **Opportunity to violate** | | | | | | | | | | | | | | | | | | | | |
| **x Month: June (vs May)** | | | | | | | | | | | | | | | | | | | .00 (.02) | .00 |
| **x Month: July (vs May)** | | | | | | | | | | | | | | | | | | | .01 (.02) | .02 |

*Note.* Robust standard errors between parentheses.

\* $p < .05$

\*\* $p < .01$

\*\*\* $p < .001$.

**Table 6. Mediation models: Total, direct, and indirect effects per mediator by survey waves.**

| | | Estimate | Bootstrapped SE | Lower 95% CI | Upper 95% CI |
|---|---|---|---|---|---|
| **Knowledge of measures** | | | | | |
| *x Month: June (vs May)* | Total effect | -.11 | .04 | -.20 | -.04 |
| | Direct effect | -.10 | .00 | -.18 | -.03 |
| | Indirect effect | -.01 | .00 | -.02 | -.01 |
| *x Month: July (vs May)* | Total effect | -.11 | .04 | -.18 | -.04 |
| | Direct effect | -.11 | .04 | -.17 | -.03 |
| | Indirect effect | -.01 | .00 | -.02 | -.00 |
| **Perceived health threat** | | | | | |
| *x Month: June (vs May)* | Total effect | -.09 | .04 | -.18 | -.02 |
| | Direct effect | -.10 | .04 | -.18 | -.03 |
| | Indirect effect | .01 | .01 | -.00 | .02 |
| *x Month: July (vs May)* | Total effect | -.07 | .04 | -.15 | .00 |
| | Direct effect | -.11 | .04 | -.17 | -.03 |
| | Indirect effect | .03 | .01 | .02 | .05 |
| **Personal costs** | | | | | |
| *x Month: June (vs May)* | Total effect | -.10 | -.04 | -.18 | -.03 |
| | Direct effect | -.10 | .04 | -.18 | -.03 |
| | Indirect effect | -.00 | .00 | -.01 | .00 |
| *x Month: July (vs May)* | Total effect | -.11 | .04 | -.18 | -.03 |
| | Direct effect | -.11 | .04 | -.17 | -.03 |
| | Indirect effect | -.00 | .00 | -.01 | .00 |
| **Moral alignment** | | | | | |
| *x Month: June (vs May)* | Total effect | -.11 | .04 | -.20 | -.04 |
| | Direct effect | -.10 | .04 | -.18 | -.03 |
| | Indirect effect | -.01 | .01 | -.02 | .01 |
| *x Month: July (vs May)* | Total effect | -.12 | .04 | -.19 | -.04 |
| | Direct effect | -.10 | .04 | -.17 | -.03 |
| | Indirect effect | -.02 | .01 | -.03 | -.00 |
| **Descriptive social norms** | | | | | |
| *x Month: June (vs May)* | Total effect | -.11 | .04 | -.19 | -.04 |
| | Direct effect | -.10 | .04 | -.18 | -.03 |
| | Indirect effect | -.01 | .00 | -.02 | -.00 |
| *x Month: July (vs May)* | Total effect | -.12 | .04 | -.18 | -.03 |
| | Direct effect | -.11 | .04 | -.17 | -.03 |
| | Indirect effect | -.01 | .00 | -.02 | -.00 |
| **Practical capacity to adhere** | | | | | |
| *x Month: June (vs May)* | Total effect | -.08 | .04 | -.16 | -.00 |
| | Direct effect | -.10 | .04 | -.18 | -.03 |
| | Indirect effect | .02 | .01 | -.00 | .05 |
| *x Month: July (vs May)* | Total effect | -.10 | .04 | -.17 | -.01 |
| | Direct effect | -.11 | .04 | -.17 | -.03 |
| | Indirect effect | .01 | .01 | -.01 | .04 |
| **Opportunity to violate** | | | | | |
| *x Month: June (vs May)* | Total effect | -.11 | .04 | -.19 | -.03 |
| | Direct effect | -.10 | .04 | -.18 | -.03 |
| | Indirect effect | -.01 | .00 | -.01 | -.00 |
| *x Month: July (vs May)* | Total effect | -.11 | .04 | -.18 | -.03 |

(*Continued*)

**Table 6.** (Continued)

| | | Estimate | Bootstrapped SE | Lower 95% CI | Upper 95% CI |
|---|---|---|---|---|---|
| | Direct effect | -.11 | .04 | -.17 | -.03 |
| | Indirect effect | -.00 | .00 | -.01 | -.00 |

lower knowledge of mitigation measures, by reductions in the perceived health threat of COVID-19, by people's lower alignment with social distancing measures, and by reduced (descriptive) social norms for keeping distance. The notion that the personal cost of mitigation measures decreased during this period, and that people became more practically capable of adhering to these, did not counter these trends.

## Discussion

The results of our study show that a broad range of behavioral mechanisms has been at play in shaping adherence to pandemic mitigation measures in the period that followed the first wave lockdown against COVID-19. In the period after stricter mitigation measures were repealed, during the summer months of 2020, a significant decline in adherence was observed. Across this period, adherence to social distancing measures was shaped by a range of factors, relating to people's practical knowledge and understanding of mitigation measures, their perceptions of their costs and benefits, their perceptions of legitimacy and procedural justice, their personal factors, their social environment, and their practical circumstances. Moreover, changes in the levels of these factors during this period explained (in part) the observed decline in adherence. These findings demonstrate that large-scale behavioral change can be accomplished through a combination of factors situated at different levels. Yet, the study also shows that some variables that have received much attention in general psychological, economic, and criminological compliance scholarship did not play a clear and consistent role in shaping adherence.

Across the different steps of our analysis, eight variables emerged as consistent predictors of adherence. Respondents adhered more when (1) they had greater knowledge of social distancing measures, (2) they perceived the virus as a more severe health threat, (3) adherence was more costly for them (possibly reflecting the reverse: that costs were higher for those who adhered more), (4) they morally agreed more with the measures, (5) they were low in impulsivity, (6) they perceived stronger (descriptive) social norms for keeping a safe distance, (7) they had greater practical ability to adhere, and (8) they perceived fewer opportunities for violating the measures. When examining their effect sizes in the final step of the regression model, however, it becomes clear that especially capacity had a critical impact on respondents' adherence (according to Cohen's standards, a large effect). Moral alignment and perceived threat also had a substantial, but smaller impact on adherence (according to Cohen's standards, a small to medium effect). The impact of impulsivity, knowledge, opportunity for violating, personal costs, and social norms was only limited, however (according to Cohen's standards, a small effect).

The impact of the predictors on adherence was largely consistent throughout this period, although the influence of impulsivity became gradually weaker (and the influence of non-normative obligation to obey and trust in science gradually stronger) as the distance from the lockdown period increased. The decline that occurred across this period in levels of knowledge, moral alignment, and perceived social norms for adhering partially explained the observed decrease in adherence. Conversely, the increase in perceived threat that was observed toward the end of this period positively affected the development of adherence. Other variables, however, failed to predict adherence, or no longer did so when other variables were

taken into account. These most notably included procedural justice [40, 41], obligation to obey the law or the responsible authorities [46, 47], deterrence [33–35], and trust in science [50, 51].

Theoretically, the present comparison of adherence over the summer months demonstrates that the nature of behavioral change and influence on behavior is not static. Rather, our findings show that across similar samples of people, with similar measures staying in place, key factors that sustain compliance can grow or decline even in a matter of months. Our data allow us to trace these processes more deeply by examining how the key predictors have changed over the summer months. Although the influence of these variables on adherence was largely consistent throughout this period, the data revealed significant changes in their absolute levels. People reported, for instance, to have more opportunity to violate the social distancing measures (which makes sense given that stay-at-home orders were mostly lifted in this period), lower capacity to adhere, and lower perceived social norms for adherence (consistent with the notion that there were larger crowds, and that more people were expected to resume normal work and social activities). Our mediation analyses revealed that the observed decline in adherence to social distancing measures that was observed during this period was partially explained by the decreases in people's knowledge, moral alignment, and perceived social norms for adhering. Conversely, the increase in perceived threat that was observed toward the end of this period positively affected the development of adherence. When viewed together, these changes provide important indications of why adherence has changed over time. These processes do not seem to indicate that there was a so-called general behavioral fatigue [91, 92] at play at this time, but rather that lower adherence may have resulted from very particular and factual changes in people's circumstances, the environment, and their motivations. By providing insight into which variables do (and do not) shape adherence, the present research offers a more practical way of assessing whether people are able to sustain behavioral change for as long as needed, compared to broad and vague concepts such as behavioral fatigue (which rely more on common-sense understanding than mechanisms from behavioral science). An important question for future research, however, is to understand more deeply how the changes that we observed across this period may be connected to local developments in policy, society, and the pandemic (e.g., see [93]). For this, a more fine-grained analysis is needed, which takes into account how these processes developed locally at the level of regions, states, counties, or even cities.

Our findings on deterrence deserve extra discussion. In light of the fact that stricter mitigation measures have been repealed, and thus are no longer widely enforced [94], it is noteworthy that Americans nevertheless reported moderately high levels (i.e., close to the scale midpoint) of perceived punishment certainty and severity. One explanation for such continuing perceptions of deterrence when there is no longer any enforcement is that there are spillover effects. In this case, this might mean that prior enforcement continues to drive deterrence perceptions even after it has ended, or that enforcement of other measures (e.g., facemasks; quarantine) also shapes deterrence perceptions for social distancing [95]. A second, and related explanation is that people generally do not have very good perceptions of deterrence and can underestimate or overestimate both the certainty and severity of punishment [96]. Importantly, however, even though many Americans considered it quite likely that they would be punished when not keeping a safe distance, and regarded such punishment as quite severe, these beliefs did not predict greater adherence. This finding is in line with studies in other countries where there was actual enforcement of social distancing measures, where also no effects of deterrence on compliance were observed [18]. However, these conclusions clearly oppose belief in the effectiveness of strong punishment for COVID-19 violations [97, 98].

Clearly, the data allow for the exploration of many other relationships beyond those that we study in the present manuscript. For example, the data can inform about relationship between

adherence and political orientation or trust in science (both singled out as important predictors of adherence in prior research [50, 51, 99, 100], yet neither a significant predictor in our final regression model), or demographic factors like ethnicity or socio-economic status. From the results of the hierarchical regression analysis, it seems plausible that these and other factors may have indirect relationships with adherence, through their effects on more proximal predictors. The data further could illuminate how specific subsets of predictors may interact with each other, or could be used to study other outcome variables (e.g., how these predictors may explain felt negative emotions, or support for authorities, etc.). The present research was primarily oriented on understanding the proximal predictors of adherence. For this reason, we feel that other relationships, such as those outlined above, are best reserved for dedicated manuscripts that are specifically oriented on these questions. We welcome further analyses of these questions, and have made our data publicly available for this purpose. Future research could also expand on these findings by zooming in further on specific variables that may directly or indirectly shape compliance (e.g., by distinguishing essential and nonessential work; by separating individuals from different generations [101]), or by identifying further variables with which our model could be expanded.

Our findings have several policy implications, which may aid authorities in the U.S. and elsewhere to sustain adherence with mitigation measures, both for the current outbreak and for future pandemics. The results of our surveys identify seven factors that influence adherence. We formulate recommendations based on the most influential of these.

First, and most critically, authorities can increase adherence by making it practically easier for citizens to do so, and by removing opportunities to offend. Indeed, in terms of effect size, people's practical capacity to adhere was the strongest predictor of adherence, by some margin. This suggests that authorities can have an important impact on adherence by increasing citizens' practical capacity to do so. In context of social distancing, this has included arrangements that guide crowds through public venues in ways that keep them apart as much as possible, facilitating telework where possible, instituting caps on the number of people able to enter a public space, and so forth. Conversely, authorities can also shape adherence by removing practical opportunities for not following mitigation measures. Such measures are best reserved for especially harmful offenses that are widely condemned, however, because if these are not widely supported, overly coercive measures may strongly undermine citizens' motivation [102].

Second, our results show that individuals adhere more when they morally agree with mitigation measures. This finding suggests that authorities can increase adherence if they can effectively convince citizens of the importance and legitimacy of such measures. In the case of social distancing, this has included presenting evidence of how social distancing measures can prevent the spread of the virus, or emphasizing citizens' shared moral duty to protect vulnerable individuals. Cultivating citizens' support–or conversely, attuning mitigation measures to what is widely supported–will increase the chance that citizens will effectively adhere to such measures.

Third, perceptions of threat to oneself and others are an important predictor of adherence to mitigation measures. However, the present findings also demonstrate that threat perceptions are dynamic. Here, threat perceptions increased from June to July, reflecting the increase in infections that occurred during this period [6]. Findings from our studies in the Netherlands [18], however, demonstrate that threat perceptions can quickly recede as infections decline, with deleterious effects on support for, and adherence to, mitigation measures. Accordingly, to sustain adherence to mitigation measures, it is important that authorities do not give the impression that the threat is waning once infections recede [103]. Rather,

authorities can sustain adherence if they successfully convince citizens of the continuing threat of the pandemic, for example to themselves or vulnerable others.

Fourth, our findings show that knowledge of mitigation measures is important for adherence. Due to the fragmented authority response in the U.S., mitigation measures may differ substantially between states, counties, and municipalities. Consequently, it can be unclear to citizens what mitigation measures require of them. Accordingly, authorities can promote adherence by clearly communicating what the measures are and what they require of citizens.

Finally, our findings demonstrate that people's adherence to mitigation measures is influenced by the behavior of others in their community (i.e., descriptive social norms). Although this effect was modest in terms of effect size in the final regression model, effects of social norms on social distancing have also been demonstrated in other research [18, 104, 105]. Authorities thus can enhance adherence to mitigation measures by demonstrating that adherence is common and widely approved of. This also means, however, that authorities should take care to not convey the impression that violations are ubiquitous and normal. It is plausible that in the period after the initial lockdown, highly publicized instances where people widely disregarded social distancing measures may have undermined adherence, by normalizing lack of distancing. To promote adherence to mitigation measures, authorities therefore should express that doing so is the norm (or ought to be), highlight examples where many others are seen to adhere, not draw undue attention to examples where people do not, and to ensure that they are always seen to adhere to the measures themselves.

Overall, the study of adherence of social distancing measures has important implications for the study of compliance generally and the way rules shape human behavior. These questions have been studied across different academic domains, and with a focus on different mechanisms and interventions [106]. This has resulted in a patchwork of theories and approaches that are seldomly brought together, which exist in compartmentalized silos that draw on their own literatures, methods and findings. The present study brings together a broad range of variables from across these approaches, situated at different levels (i.e., the individual, the social environment, the practical circumstances), and reveals how these together shape adherence in context of social distancing measures. Although the associations that were observed here may not extend beyond this setting, the insight that adherence derived from such a diverse range of influences is nevertheless important for study of compliance. It underlines that to better understand why people comply, research can benefit from a multi-theoretical approach, in which the extant, siloed literatures are brought together and integrated.

Our study has several limitations. First, although our samples were large and stratified sampled by age, gender, and ethnicity to mimic the demographic characteristics of the United States population based on U.S. Census Bureau data, they remain non-probability convenience samples. Furthermore, there was some variability between the samples in terms of demographics, possibly due to the considerable subset of participants who failed to complete the survey or pass the attention checks. As a consequence, our samples cannot be regarded as truly nationally representative. Nevertheless, there is evidence that such convenience samples can be as accurate as random digit dial telephone surveys [107, 108], and they may reduce social desirability biases [109]. Further research therefore is needed to understand the robustness of the observed findings, although they align with evidence from other research [3]. Second, our surveys rely on self-reported measures that may be subject to imperfect recall or social desirability bias [110, 111]. We do note, however, that a recent study demonstrated that social desirability bias did not inflate the estimates of compliance with COVID-19 measures in online surveys [112], and that the finding of high self-reported adherence is in line with objective data from Google COVID19 Community Mobility [113]. Furthermore, prior research shows that there can be strong concordance between self-reported and objective compliance measures when

surveys are used (see [78] p. 29). Even so, future research into these questions would benefit from methods that supplement self-reported measures with behavioral data, such as video observation [114].

## Conclusion

In the summer of 2020, the Federal lockdown and stay-at-home measures against COVID-19 that were in force in spring were lifted, and in large parts of the country, society began to reopen. The present findings, based on three stratified samples collected in May, June, and July, show that Americans' adherence to social distancing measures declined, as did several of the factors that sustained it–including people's practical capacity to adhere, their knowledge of the measures, and social norms for adherence. Our research identifies key variables that predicted greater adherence as society reopened, and which contributed to the changes in adherence that were observed thereafter. By doing so, this research contributes to the understanding of pandemic governance and the interaction between rules and human conduct more generally. Moreover, in the current stage of the pandemic, these findings provide important directions for the public health response, by highlighting processes through which adherence to mitigation measures can be promoted, as we strive to return to normality.

## Supporting information

**S1 Survey. Survey materials.**
(PDF)

**S1 Dataset. Dataset and syntax files.**
(DOCX)

**S1 Table. Kendall's tau correlations between demographic variables and adherence.** May 8–18 (Survey 1. N = 1012). Note. *–Correlation is significant at the .05 level. **–Correlation is significant at the .01 level. Gender–Female as reference category. Political orientation– N = 866.
(DOCX)

**S2 Table. Kendall's tau correlations between demographic variables and adherence.** June 8–16 (Survey 2. N = 986). Note. *–Correlation is significant at the .05 level. **–Correlation is significant at the .01 level. Gender–Female as reference category. Political orientation– N = 880.
(DOCX)

**S3 Table. Kendall's tau correlations between demographic variables and adherence.** July 11–17 (Survey 3. N = 921). Note. *–Correlation is significant at the .05 level. **–Correlation is significant at the .01 level. Gender–Female as reference category. Political orientation– N = 803.
(DOCX)

**S4 Table. Kendall's tau correlations between independent variables and adherence.** May 8–18 (Survey 1. N = 1012). Note. *–Correlation is significant at the .05 level. **–Correlation is significant at the .01 level.
(DOCX)

**S5 Table. Kendall's tau correlations between independent variables and adherence.** June 8–16 (Survey 2. N = 986). Note. *–Correlation is significant at the .05 level. **–Correlation is

significant at the .01 level.
(DOCX)

**S6 Table. Kendall's tau correlations between independent variables and adherence.** July 11–17 (Survey 3. N = 921). Note. *–Correlation is significant at the .05 level. **–Correlation is significant at the .01 level.
(DOCX)

**S1 Output. Complete regression output Survey 1 (May).**
(PDF)

**S2 Output. Complete regression output Survey 2 (June).**
(PDF)

**S3 Output. Complete regression output Survey 3 (July).**
(PDF)

**S4 Output. Complete regression output Surveys 1–3 combined (May-July).**
(PDF)

## Author Contributions

**Conceptualization:** Christopher P. Reinders Folmer, Benjamin van Rooij.

**Data curation:** Christopher P. Reinders Folmer.

**Formal analysis:** Christopher P. Reinders Folmer.

**Funding acquisition:** Benjamin van Rooij.

**Investigation:** Christopher P. Reinders Folmer, Megan A. Brownlee.

**Methodology:** Christopher P. Reinders Folmer, Megan A. Brownlee, Adam D. Fine, Emmeke B. Kooistra, Malouke E. Kuiper, Elke H. Olthuis, Anne Leonore de Bruijn, Benjamin van Rooij.

**Project administration:** Christopher P. Reinders Folmer.

**Resources:** Christopher P. Reinders Folmer, Adam D. Fine, Emmeke B. Kooistra, Malouke E. Kuiper, Elke H. Olthuis, Anne Leonore de Bruijn, Benjamin van Rooij.

**Software:** Christopher P. Reinders Folmer, Megan A. Brownlee.

**Supervision:** Benjamin van Rooij.

**Visualization:** Christopher P. Reinders Folmer.

**Writing – original draft:** Christopher P. Reinders Folmer, Megan A. Brownlee, Adam D. Fine, Benjamin van Rooij.

**Writing – review & editing:** Christopher P. Reinders Folmer, Megan A. Brownlee, Adam D. Fine, Emmeke B. Kooistra, Malouke E. Kuiper, Elke H. Olthuis, Anne Leonore de Bruijn, Benjamin van Rooij.

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
