## [Decision Letter · Decision Letter 0]

5 Jun 2021

PONE-D-20-39307

Social Distancing in America 

Understanding Long-term Adherence to COVID-19 Mitigation Recommendations

PLOS ONE

Dear Dr. Reinders Folmer,

Thank you for submitting your manuscript to PLOS ONE. After careful consideration, we feel that it has merit but does not fully meet PLOS ONE’s publication criteria as it currently stands. Therefore, we invite you to submit a revised version of the manuscript that addresses the points raised during the review process.

We look forward to receiving your revised manuscript.

Kind regards,

Ali B. Mahmoud, Ph.D.

Academic Editor

PLOS ONE

Journal Requirements:

Please include additional information regarding the survey or questionnaire used in the study and ensure that you have provided sufficient details that others could replicate the analyses. For instance, if you developed a questionnaire as part of this study and it is not under a copyright more restrictive than CC-BY, please include a copy, in both the original language and English, as Supporting Information. Moreover, please include more details on how the questionnaire was pre-tested, and whether it was validated.

Reviewers' comments:

Reviewer's Responses to Questions

**Comments to the Author**

1. Is the manuscript technically sound, and do the data support the conclusions?

Reviewer #1: Partly

Reviewer #2: Yes

2. Has the statistical analysis been performed appropriately and rigorously? 

Reviewer #1: No

Reviewer #2: Yes

3. Have the authors made all data underlying the findings in their manuscript fully available?

Reviewer #1: Yes

Reviewer #2: Yes

4. Is the manuscript presented in an intelligible fashion and written in standard English?

Reviewer #1: Yes

Reviewer #2: Yes

5. Review Comments to the Author

Reviewer #1: This paper presents findings from three cross-sectional samples of US residents in May, June and July of 2020 to determine factors that are related to social distancing behavior during and after the lockdown in many states in the US. The surveys covered a great deal of potential influences on that behavior and seeing how those factors changed in their relations with social distancing behavior is interesting and informative. I do have some suggestions to make the paper easier to digest, because there are so many analyses, one is likely to get lost in all of the approaches taken.

In regard to the main analysis, I don’t see why this is divided into two separate models as shown in Tables 4 and 5. The model in Table 5 seems quite different from the one in Table 4 and I didn’t understand why the variables in Table 4 were treated differently from the ones in Table 5. Why can’t all of this be done in one model?

In addition, it would be valuable to conduct the analyses in a more hierarchical manner, so that demographic and other personal characteristics were entered first and the various other types of variables were entered in blocks as you have done in Table 5. One can then see how various beliefs might be associated with the variables entered first, such as political ideology. As you note, this characteristic was not significant in the model when all variables were entered simultaneously, but it was when it was entered before a lot of the other beliefs. Another way to handle this is to present a table with all of the variables tested for their univariate relationship with social distancing. But this would not allow one to see how relations change as new variables are added to the model. I also am puzzled as to why a variable like trust in science or media is treated like a control variable. These are very important considerations in whether someone will adhere to government recommendations. I would place them together with other beliefs such as those regarding respect for the law.

I am also a bit puzzled by the measure of ability to practice social distancing. It seems to be very similar to the actual dependent variable, and so controlling for it seems to be redundant with the outcome. If you want to use it, I would enter it later to see how it changes the earlier associations.

Given that this is a study of the US, it would also be helpful to provide a breakdown of the geographic distribution of the sample. One could use the four census divisions as a way to do that. These divisions are also of interest because there was considerable variation in compliance with social distancing recommendations in different parts of the US. This could go into the first set of predictors in a hierarchical model.

I think the national representativeness of the sample should be downplayed in the description of it in the Method. This is basically a convenience sample that was recruited with demographic targets aimed to be representative of the US. But that is not necessarily probability-based, and so the conclusions that one can draw must be tempered to a degree.

I am also puzzled by the definition of situational variables. I don’t see how impulsivity is situational. This is a personality disposition that is relatively stable. Negative emotions are not necessarily situational, especially during a crisis like a pandemic. Knowledge and understanding of measures are important but they are no more situational than the perceived health threat, punishment severity, or many of the other variables in your model. Can you find a better way to organize these predictors?

I think the inclusion of criminology predictors is interesting. But in all honesty, I don’t think people regarded the social distancing mandates apart from the lockdowns as all that subject to sanctions. People were encouraged to maintain distance, but very few were arrested.

I think the abstract needs some work. No one will understand what you mean by motivational versus situational influences. The sentence that says: “as the core variables that sustain can change…” needs attention.

Reviewer #2: The research questions are clearly defined in the abstract and introduction, and are, of course, both timely and relevant. The questions and topic are interesting, especially given the protracted nature of the current pandemic, and the possibility of future pandemics. The paper is well written and easy to read. Variables are defined and explained in terms of examples nicely. In addition, the paper is well laid out and tables/figures are clear and easy to read; they aid in understanding the data and information being presented.

I found it interesting that the introduction to the paper specifically highlighted that pandemic measures such as lockdowns were eased in Southern and Midwestern parts of the US beginning in April. Given that some states, especially in the east, had much longer lockdown periods or strict pandemic rules, would location of participants not have perhaps played a role in some of the survey responses and attitudes? Additionally, there is some demographic information, such as age, ethnicity, or field of employment that may have large impacts on survey responses. For example, the study took into account those who worked directly with COVID patients/care, but other sectors of employment (ie/ the essential workers) may have also held different attitudes which may have impacted responses as well. Also, the authors mention that the study sample is nationally representative, but then highlight that that is in terms of only sex (Male, Female, Binary) or Age. Again, is the study representative in terms of location (eg/ higher populated states, areas of the country) or ethnicity? Of course, it is impossible to take every variable into account, but mention of some of these things (eg/ why they were included or excluded) might be beneficial.

6. PLOS authors have the option to publish the peer review history of their article (what does this mean?). If published, this will include your full peer review and any attached files.

Reviewer #1: **Yes: **Dan Romer

Reviewer #2: No

---

## [Author Response · Author response to Decision Letter 0]

9 Jul 2021

Editor’s comments

1. PLOS ONE style requirements

You request to ensure that the manuscript meets PLOS ONE’s style requirements, including those for file naming.

We thank you for reminding us of this. We have carefully re-checked this to ensure that the manuscript meets all style requirements, including those for file naming.

2. Information about the survey and the analyses

You ask that additional information is provided about the survey that was used in the study, and to ensure that sufficient details are presented to allow others to replicate the analyses.

We agree that it is important to ensure that sufficient information is provided about the survey and the analyses. In response, we have included all survey materials in the revised submission as Supplementary Information. Furthermore, we discuss in more detail the steps that were taken in developing the survey (p. 13):

“Our survey (see Supporting Information) was based on our prior surveys conducted in April 2020 in the United States [5], the United Kingdom [77], the Netherlands [78], and Israel [79]. It assessed the same variables and relied on the same measures. Measures that displayed poor internal consistency in the previous surveys were revised to improve their internal consistency (e.g., adherence, social norms, capacity to adhere, and opportunity to violate); reliability of the revised measures was high (α ≥ .85, more details below).”

Furthermore, the Supplementary Information included with the manuscripts includes all syntax required to replicate the analyses. These are presented in structured fashion so that all analyses can directly be replicated by running them. 

Reviewer 1’s comments

1. Regression model

The Reviewer wonders why the main analysis was divided into two regression models, rather than a single model. The Reviewer suggests to present the analysis as a single model, in which different types of variables are added in blocks.

We thank the Reviewer for this suggestion. We agree that conducting the analyses as a hierarchical regression model provides greater clarity, for example into how the relationships shown in previous blocks are affected by the inclusion of further predictors in subsequent blocks. In the revised manuscript, we have adopted this approach (see p. 28-35). We conduct the regression analysis as a hierarchical model, starting with the control variables, and adding six categories of variables in sequential steps: (1) variables relating to people’s practical knowledge and understanding of the mitigation measures, (2) variables relating to their perceptions of their costs and benefits, (3) variables relating to their evaluation and felt obligation toward the measures and the responsible authorities, (4) variables relating to personal factors, (5) variables relating to their social environment, and (6) variables relating to their practical circumstances. 

To further increase the clarity and parsimoniousness of these analyses, the separate analyses that were conducted per wave in the previous version of the manuscript have been replaced by a single regression model, based on the aggregated data. This model captures developments in adherence across survey waves by including survey wave as a predictor. Furthermore, to understand how the impact of the predictors has changed across this period, it further explores interactions between the predictors and survey wave. Finally, the analyses explore how the observed developments in the predictors (i.e., the changes in their absolute levels across this period) have contributed to the decline in adherence. For this purpose, we have added mediation analyses. 

In sum, based on the Reviewer’s recommendations, we have thoroughly revised the analyses. The revised analyses improve upon the original version by illuminating (A) which variables predict greater adherence, (B) how their impact is affected by the inclusion of other variables in subsequent steps, (C) how their impact has changed across survey waves, and (D) how developments in the predictors explain the observed decline in adherence. 

2. Trust in science and media

The Reviewer wonders why trust in science and trust in media were treated as control variables, in light of the important impact that such considerations may have on adherence to government recommendations. 

The Reviewer is right that trust in science and media can be highly relevant for adherence. In response to the recommendations, we include these factors in our predictors in the revised classification of our predictors (see Reviewer 1’s points 1 and 6). We have arranged them under personal factors, which are added in Step 5 of the hierarchical regression model.

3. Capacity to adhere

The Reviewer observes that capacity to adhere appears to be very similar to the actual dependent variable, and wonders whether it may be redundant. If not, the Reviewer advises to enter it later in the model, to demonstrate how it changes the earlier associations.

Indeed, people’s capacity to adhere obviously can have an important impact on their actual adherence. Yet it is important to note that conceptually, these are very different variables. Capacity captures the notion that the circumstances may make it more or less difficult for individuals to adhere, such that people must exert more (or less) effort to effectively do so. This does not imply that high or low capacity automatically result in (non)adherence, however. Simply having the capacity to commit a crime, does not mean that one also will do so. Indeed, for many types of offenses, people’s capacity to offend may generally be high (e.g., theft or murder), yet few individuals effectively commit such transgressions. The same applies to social distancing: being practically able to keep a distance from others does not mean that someone always wishes to do so. For this reason, capacity to adhere remains an important, conceptually distinct variable. To assuage the Reviewers’ concerns, we now include it in the final step of the regression model, so that its impact on the preceding steps can readily be observed (see p. 30-33; relative to the preceding steps, only the effects of normative obligation to obey and trust in science were rendered nonsignificant). Furthermore, we discuss the relationship between capacity and adherence in greater detail in the introduction (see p. 8-9):

“In order for people to effectively do as social distancing measures demand, it is necessary that their practical circumstances effectively allow them to do so. However, in practice, their capacity to adhere may often vary. For example, keeping a safe distance from others may be more difficult in crowded or constrained environments, or in occupations that cannot be conducted from home or at a distance. Capacity thus may strongly shape adherence, but it should be understood that these concepts are not identical. Simply having the capacity to commit a crime does not mean that one also will do so. The same applies to social distancing: being practically able to keep a distance from others does not mean that someone wishes to do so. We expected adherence with social distancing measures to be higher among people who had greater practical capacity to adhere to these measures.”

4. Geographic distribution

The Reviewer observes that it would be helpful to provide a breakdown of the geographic distribution of the sample, for example based on the four census divisions. The Reviewer notes that this could be included to the control variables in the first step of the regression model.

We thank the Reviewer for this suggestion, and have followed their recommendation. We now present the geographic distribution of our samples by census region (see Table 1, p. 11-12), and include region (dummy-coded) as a control variable in the first step of the regression model (Table 4, p. 30-33). Relative to respondents from the Northeast region, adherence was significantly lower among respondents from the Midwest and the South regions. However, it is important to note that these differences were mostly rendered nonsignificant by the addition of further predictors in subsequent steps of our model, such as moral alignment and normative obligation to obey (Step 4) and descriptive social norms (Step 6). 

5. National representativeness of the sample

The Reviewer remarks that the national representativeness of the sample should be downplayed in the Method section, given that it basically is a convenience sample that was recruited with demographic targets aimed to be representative of the US.

The Reviewer is right that our sample was stratified according to demographic targets based on US Census stratifications. In the revised manuscript, we have specified this more clearly, and have toned down all claims that the sample is nationally representative (e.g., in the abstract, introduction, method, discussion). We also explicate our sampling strategy more clearly in the method section (p. 10):

“Participants were residents (18 years or older) of the U.S. that were recruited via the online survey platform SurveyMonkey (https://surveymonkey.com). They were recruited using a stratified sampling approach, in which the final intended sample size was divided into subgroups with the same demographic proportions (age, gender, and race/ethnicity) as the national population based on estimates from the U.S. Census Bureau (https://www.census.gov/). This stratified sampling approach mimics the demographic characteristics of the United States, though it retains the biases and characteristics of a non-probability convenience sample.”

Moreover, we mention this as a limitation of our study in the Discussion (p. 52):

“First, although our samples were large and stratified sampled by age, gender, and ethnicity to mimic the demographic characteristics of the United States population based on U.S. Census Bureau data, they remain non-probability convenience samples. Furthermore, there was some variability between the samples in terms of demographics, possibly due to the considerable subset of participants who failed to complete the survey or pass the attention checks. As a consequence, our samples cannot be regarded as truly nationally representative. Nevertheless, there is evidence that such convenience samples can be as accurate as random digit dial telephone surveys [105, 106], and they may reduce social desirability biases [107].”

6. Organization of predictors

The Reviewer wondered about the classification of the predictors, especially the ‘situational’ variables, and asks if we can find a better way to organize these. 

We thank the Reviewer for this suggestion. In the revised manuscript, we present a revised classification of the predictors. In this typology, the predictors are classified into six categories: (1) variables relating to people’s practical knowledge and understanding of the mitigation measures, (2) variables relating to their perceptions of their costs and benefits, (3) variables relating to their perceptions of legitimacy, procedural justice, and their obligation to obey the measures and the responsible authorities, (4) variables relating to personal factors, (5) variables relating to their social environment, and (6) variables relating to their practical circumstances. We have also augmented our substantive explanation of these categories on p. 5-9 of the revised manuscript. To show their respective contribution to adherence, these categories are entered in sequential steps in the hierarchical regression analysis (see Reviewer 1’s point 1). We believe that this approach better illuminates how different types of variables contribute to adherence, and hope that the Reviewer agrees with us. 

7. Sanctions

The Reviewer remarks that although the inclusion of criminological predictors was interesting, people probably did not regard social distancing measures as very subject to sanctions, as very few people were effectively arrested.

Our decision to include deterrence was motivated by the fact that in general, punishment is a major intervention that is used to promote compliance, and also a major theoretical approach, based in rational choice. Furthermore, as the Reviewer observes, punishment did occur during the lockdown phase, and moreover, strong sanctions were threatened or imposed for other violations of mitigation measures (e.g., threats of prosecuting coughing attacks as terrorism). Research on deterrence shows that it is particularly perceptions of punishment certainty and severity, rather than the actual chance or severity of punishment, that is important for compliance (e.g., [32]; also see Apel, 2013; Decker et al., 1993). In light of prior enforcement, and continuing enforcement of other measures related to COVID-19, it is possible that respondents’ perceptions of punishment certainty and severity may still be meaningful, even when enforcement is trivial in practice. Indeed, it is noteworthy that respondents nevertheless reported moderately high levels (i.e., close to the scale midpoint) of perceived punishment certainty and severity, and that considerable variability existed in these perceptions – such that some people regarded punishment as more certain and meaningful than others (see Table 3, p. 21-22). Accordingly, we were interested in how such perceptions may shape adherence. In the revised manuscript, we outline this reasoning in the introduction (see p. 6):

“Although social distancing measures were not widely enforced in the U.S., sanctions did occur during the first wave lockdown [30]; furthermore, severe sanctions were communicated for other COVID-related violations [31]. Research on perceptual deterrence suggests that subjective perceptions of punishment may also influence compliance [32]. For this reason, we also examined subjective perceptions of punishment for not following social distancing measures, separating punishment certainty and severity – the key dimensions separated by general deterrence theory [33-35].”

8. Abstract

The Reviewer points out that the abstract needs work as readers will not understand ‘situational’ and ‘motivational’ variables without further specification.

We thank the Reviewer for mentioning this. In response, we have thoroughly revised the abstract, to correspond with the revised classification of variables. In the revised abstract, we now mention the six categories of variables, as well as the specific variables that were found to predict adherence. Furthermore, in the revised abstract, we also mention the results of the mediation analysis, in terms of the variables that were found to explain the observed decline in compliance across survey waves (p. 2): 

“(…) For this purpose, we examined a broad range of factors, relating to people’s (1) knowledge and understanding of the mitigation measures, (2) perceptions of their costs and benefits, (3) perceptions of legitimacy and procedural justice, (4) personal factors, (5) social environment, and (6) practical circumstances. Our findings reveal that adherence was chiefly shaped by three major factors: respondents adhered more when they (a) had greater practical capacity to adhere, (b) morally agreed more with the measures, and (c) perceived the virus as a more severe health threat. Adherence was shaped to a lesser extent by impulsivity, knowledge of social distancing measures, opportunities for violating, personal costs, and descriptive social norms. The results also reveal, however, that adherence declined across this period, which was partly explained by changes in people’s moral alignment, threat perceptions, knowledge, and perceived social norms. These findings show that adherence originates from a broad range of factors, which develop dynamically across time. (…).” 

Reviewer 2’s comments

1. Effect of location

The Reviewer wonders whether participants’ location may not have played a role in their responses, given that some states were later to repeal lockdown measures than others.

We thank the Reviewer for this question, which is similar to a point raised by Reviewer 1 (point 4). Indeed, it is likely that geographic region may have influenced participants’ responses, given that differences between regions existed in terms of the length of lockdown measures, as well as levels of infections. In the revised manuscript, we now include region (based on the U.S. census regions) as an additional control variable in our regression model. As stated in our response to Reviewer 1, adherence was lower in the Midwest and South regions than in the Northeast. However, such regional differences were mostly reduced to nonsignificance when other, more proximal predictors were added to the model.

2. Influence of demographic and other variables

The Reviewer observes that there is some demographic information that may have had an important impact on survey responses. The Reviewer specifically mentions age, gender and ethnicity, and field of employment (particularly essential work).

The Reviewer is correct that many demographic variables may have shaped responses to the survey. For this reason, we controlled for these variables in all analyses. We do recognize, however, that many variables might have important indirect effects on adherence, for example by shaping key predictors (e.g., perceived threat, moral alignment, capacity to comply). These may include the demographic variables mentioned by the Reviewer, but also region (see Reviewer 1 point 4 and Reviewer 2 point 1), political orientation, or trust in science and media (see Reviewer 1’s point 2). While we regard such indirect relationships as highly interesting, we do feel that they are too many to analyze in sufficient detail in the present manuscript, which is oriented on understanding more proximal, direct relationships with adherence. However, we do wholeheartedly encourage other researchers to explore further indirect and interactive relationships using these data, and have made them publicly available for this purpose. We outline this reasoning in the discussion (see p. 49-50):

“Clearly, the data allow for the exploration of many other relationships beyond those that we study in the present manuscript. For example, the data can inform about relationship between adherence and political orientation or trust in science (both singled out as important predictors of adherence in prior research [50, 51, 98, 99], yet neither a significant predictor in our final regression model), or demographic factors like ethnicity or socio-economic status. From the results of the hierarchical regression analysis, it seems plausible that these and other factors may have indirect relationships with adherence, through their effects on more proximal predictors. The data further could illuminate how specific subsets of predictors may interact with each other, or could be used to study other outcome variables (e.g., how these predictors may explain felt negative emotions, or support for authorities, etc.). The present research was primarily oriented on understanding the proximal predictors of adherence. For this reason, we feel that other relationships, such as those outlined above, are best reserved for dedicated manuscripts that are specifically oriented on these questions. We welcome further analyses of these questions, and have made our data publicly available for this purpose.”

We agree with the Reviewer that the distinction between essential and nonessential work is important. Regrettably, we did not measure this in the present research, and thus could not explore this further.

3. National representativeness of the sample

The Reviewer wonders to what extent our sample was nationally representative on other features than sex and age, such as location and ethnicity.

This point is related to that raised by Reviewer 1 (point 5), and we have responded in detail there. In brief, our participants were stratified sampled into pre-determined subgroups with the same demographic proportions (age, gender, and race/ethnicity) as the national population based on U.S. Census Bureau statistics. As such, it comes very close to being nationally representative on demographics. However, the result remains a non-probability convenience sample, rather than a truly nationally representative sample (if one truly exists). We have explicated this more clearly in the method section (p. 10), have acknowledged this as a limitation in the discussion (p. 52), and have toned down all claims of national representativeness from the manuscript. We did not directly measure ethnic group membership (only whether respondents regarded themselves as part of a minority group), and therefore cannot directly compare these to national statistics from the census (though the sample was stratified to have the same proportions as the national population). 

 Our sample was not designed to be stratified according to location. A comparison with the estimates from the U.S. Census Bureau indicates that our sample somewhat overrepresented residents from the Northeast (20.4%, vs. 17.2% in the census) and South (42.8%, vs. 38.1%), underrepresented residents of the West (16.0%, vs. 23.8%), and was almost equivalent in residents from the Midwest (20.8%, vs. 20.9%). To fully examine any regional effects, we would need a substantially larger sample, though it is important to note that the regional differences we uncovered were both small in magnitude and generally rendered non-significant when other variables were included. We mention in the Discussion that further research is needed to understand how these processes occurred at the regional or subordinate levels (p. 48):

“An important question for future research, however, is to understand more deeply how the changes that we observed across this period may be connected to local developments in policy, society, and the pandemic. For this, a more fine-grained analysis is needed, which takes into account how these processes developed locally at the level of regions, states, counties, or even cities.”

References

Apel, R. (2013). Sanctions, perceptions, and crime: Implications for criminal deterrence. Journal of 

quantitative criminology, 29(1), 67-101. https://doi.org/10.1007/s10940-012-9170-1

Decker, S., Wright, R., & Logie, R. (1993). Perceptual deterrence among active residential burglars: 

A research note. Criminology, 31(1), 135-147. https://doi.org/10.1111/j.1745-9125.1993.tb01125.x

---

## [Decision Letter · Decision Letter 1]

13 Aug 2021

PONE-D-20-39307R1

Social Distancing in America: Understanding Long-term Adherence to COVID-19 Mitigation Recommendations

PLOS ONE

Dear Dr. Reinders Folmer,

Thank you for submitting your manuscript to PLOS ONE. The reviewers have recommended publication, but also suggested a couple of minor additions that would help improve the quality of your research. Mainly, rather than as a linear predictor, presenting the findings for age by the typical categories used for this variable. In addition, this variable may have nonlinearities that would be interesting to investigate. Further, whilst you note that the contrast between essential and non-essential work is significant; however, your study did not measure this distinction. Thus, it would be a good idea to include this in your Discussion as a research implication. Therefore, I invite you to submit a revised version of the manuscript that addresses the points raised during the review process.

We look forward to receiving your revised manuscript.

Kind regards,

Ali B. Mahmoud, Ph.D.

Academic Editor

PLOS ONE

Journal Requirements:

Reviewers' comments:

Reviewer's Responses to Questions

**Comments to the Author**

1. If the authors have adequately addressed your comments raised in a previous round of review and you feel that this manuscript is now acceptable for publication, you may indicate that here to bypass the “Comments to the Author” section, enter your conflict of interest statement in the “Confidential to Editor” section, and submit your "Accept" recommendation.

Reviewer #1: All comments have been addressed

Reviewer #2: All comments have been addressed

2. Is the manuscript technically sound, and do the data support the conclusions?

Reviewer #1: Yes

Reviewer #2: (No Response)

3. Has the statistical analysis been performed appropriately and rigorously? 

Reviewer #1: Yes

Reviewer #2: (No Response)

4. Have the authors made all data underlying the findings in their manuscript fully available?

Reviewer #1: Yes

Reviewer #2: (No Response)

5. Is the manuscript presented in an intelligible fashion and written in standard English?

Reviewer #1: Yes

Reviewer #2: (No Response)

6. Review Comments to the Author

Reviewer #1: Thank you for the careful revisions to the suggestions posed after the first submission. If I had one more suggestion, it would be to display the results for age by the typical categories used for this variable rather than as a linear predictor. There may be nonlinearities in this variable that would be interesting to see. But that is only a suggestion.

Reviewer #2: Thank you for addressing the comments made by both reviewers. The abstract is much more clear and the revised classification of the predictors, as well as the explanations for them, was well done.

One small suggestion - you mention that the distinction between essential and nonessential work is important, but was not measured in your study. Perhaps it would be prudent to add this to your Discussion, as a direction or suggestion for future research.

7. PLOS authors have the option to publish the peer review history of their article (what does this mean?). If published, this will include your full peer review and any attached files.

Reviewer #1: **Yes: **Dan Romer

Reviewer #2: No

---

## [Author Response · Author response to Decision Letter 1]

23 Aug 2021

Editor’s comments

1. Age as a linear predictor

In line with Reviewer 1’s comment, you note that age may have nonlinearities that would be interesting to investigate. As suggested by Reviewer 1, you suggest to present the findings for age by category, using the typical categories for this variable. 

We thank you and Reviewer 1 for this suggestion. We agree that age could have interesting nonlinearities that are obscured by treating this covariate as a linear variable. In response, we have carefully considered this suggestion, and conducted additional analyses to assess it. However, this has lead us to conclude that this change will not yield additional insight beyond the original analyses. 

 A first important consideration is that reducing age to a categorical variable will result in a loss of information by aggregating individuals with different ages into the same category. We also note that in similar articles in your journal, both categorical and continuous measures of age have been used.

 More importantly, when dividing participants into age categories (15-17 | 18-20 | 21-44 | 45-64 | 65+, in line with the US Census), we found no differences between age groups – neither when using the lowest age group as the reference category, nor when using the oldest age group. Moreover, using age categories did not change the results of the predictor variables (please see S5 Output). In sum: we found no indications of nonlinearities in terms of the effect of age (probably because such differences are better explained by the predictor variables, eg threat perceptions, than by the unique effect of age). Furthermore, to implement age categories in the manuscript, all analyses presented in the Results section (p. 23-46) would need to be revised (because all coefficients are slightly altered by the loss in degrees of freedom). Given that age is only a control variable in our model, and that the Reviewer presents this as “only a suggestion”, we were hesitant to make substantial changes that do not yield greater informativeness. As such, we have retained the original analyses (with age as a linear covariate) in the final manuscript. However, in case that you or the Reviewers believe that separating age by category nonetheless can be informative, we will of course be happy to revise the results accordingly. 

 We mention generational differences as a possible avenue for future research in the Discussion (p. 50), in line with your suggestion:

“Future research could also expand on these findings by zooming in further on specific variables that may directly or indirectly shape compliance (e.g., (…) by separating individuals from different generations [101])”

2. Distinction between essential and nonessential work

Following Reviewer 2’s suggestion, you suggest that it would be a good idea to include the distinction between essential and nonessential work in the Discussion as a research implication. 

We agree that this distinction would be interesting to explore, and have mentioned this as a possible avenue for future research in the Discussion (p. 50):

“Future research could also expand on these findings by zooming in further on specific variables that may directly or indirectly shape compliance (e.g., by distinguishing essential and nonessential work; by separating individuals from different generations [101]), or by identifying further variables with which our model could be expanded.”

3. Reference list

As per journal policy, you ask that we review our reference list to ensure that it is complete and correct, and to include the rationale in the case that retracted papers are cited. 

We have reviewed all references and made changes wherever appropriate (please see p. 53-64). This chiefly involved adding DOIs where these were not yet provided [19, 24-26, 32, 35, 37, 44, 55, 56, 58, 59, 61-69, 71-73, 75, 77, 79, 84, 96, 11]. Page numbers were added for [25] and [66]. A missing issue number was added for [72]. 

In addition, the previous version of the manuscript mentioned an incorrect reference for [33]; this has been replaced with the correct reference. An additional relevant reference was added to the discussion [93]. A typo was corrected for [97], and for [99] and [112] the citations were updated (from preprints to published papers). No retracted papers are cited. 

Reviewer 1’s comments

1. Age as a linear predictor

As a final suggestion, the Reviewer suggests to present the findings for age by category, using the typical categories for this variable.

We thank the Reviewer for this suggestion. We have responded to it in detail in our response to the Editor’s point 1 (please see above). 

Reviewer 2’s comments

1. Distinction between essential and nonessential work

The Reviewer notes that because we mentioned that the distinction between essential and nonessential work is important (in our rebuttal letter for the previous revision round), it might be prudent to add this to the Discussion, as a direction or suggestion for future research.

We thank the Reviewer for this suggestion. As detailed in our response to the Editor’s point 2, we now mention this in the Discussion (please see p. 50).

---

## [Editor Report · Decision Letter 2]

15 Sep 2021

Social Distancing in America: Understanding Long-term Adherence to COVID-19 Mitigation Recommendations

PONE-D-20-39307R2

Dear Dr. Reinders Folmer,

We’re pleased to inform you that your manuscript has been judged scientifically suitable for publication and will be formally accepted for publication once it meets all outstanding technical requirements.

Kind regards,

Wen-Jun Tu

Academic Editor

PLOS ONE
---

## [Editor Report · Acceptance letter]

17 Sep 2021

PONE-D-20-39307R2 

Social Distancing in America: Understanding Long-term Adherence to COVID-19 Mitigation Recommendations 

Dear Dr. Reinders Folmer:

I'm pleased to inform you that your manuscript has been deemed suitable for publication in PLOS ONE. Congratulations! Your manuscript is now with our production department. 

Kind regards, 

on behalf of

Dr. Wen-Jun Tu 

Academic Editor

PLOS ONE